# ATTENTIONAL META-LEARNERS FOR FEW-SHOT POLYTHETIC CLASSIFICATION

## ABSTRACT

Polythetic classifications, based on shared patterns of features that need neither be universal nor constant among members of a class, are common in the natural world and greatly outnumber monothetic classifications over a set of features. We show that threshold meta-learners, such as Prototypical Networks, require an embedding dimension that is exponential in the number of task-relevant features to emulate these functions. In contrast, attentional classifiers, such as Matching Networks, are polythetic by default and able to solve these problems with a linear embedding dimension. However, we find that in the presence of task-irrelevant features, inherent to meta-learning problems, attentional models are susceptible to misclassification. To address this challenge, we propose a self-attention feature-selection mechanism that adaptively dilutes non-discriminative features. We demonstrate the effectiveness of our approach in meta-learning Boolean functions, and synthetic and real-world few-shot learning tasks.

## 1 INTRODUCTION

Classification meta-learning is typically approached from the perspective of few-shot learning: Can we train a model that generalises to unseen 'natural' classes at test time? For example, in the Omniglot task (Lake et al., 2011) we have access to a labelled set of handwritten characters during training and we are tasked with distinguishing new characters, from unseen writing systems, at test time. From this perspective each example is associated with a consistent class and members of that class share a common set of properties (e.g. all handwritten characters have the shape of the underlying character class). Alternatively, we may consider meta-learning over *unseen ways of categorising*: Can we train a model on character recognition that generalises to alphabet recognition? or to distinguishing upper from lower case letters? In this setting, features need to be understood in relation to a given classification: For instance, when tasked with distinguishing equids (horses, zebras, donkeys) from big cats, the presence of stripes on both zebra and tigers is irrelevant, and potentially misleading. On the other hand, stripes are the key to distinguishing horses from zebra.

Understanding features in the context of a classification is central to the concepts of monothetic and polythetic classes recognised in the fields of taxonomy and knowledge organisation. *Monothetic classifications* are based on universal attributes: there is at least one necessary and sufficient attribute for class membership. *Polythetic classifications* are instead based on combinations of attributes, none of which are sufficient in isolation to indicate membership and, potentially, none of which are necessary. Carl Linneaus, inventor of the binomial nomenclature for species and "father of taxonomy," recognised that natural orders could not be defined monothetically, lacking features that were unique and constant over families and that, until such features could be found, such classifications were necessarily polythetic (Hjørland, 2017; Stevens, 1998).

Figure 1 illustrates Linnaeus' system for classifying plants, which relies on polythetic classifications and is still in use today. Consider, for example, that we can distinguish $\mathbb{A}$ from $\mathbb{B}$ by the number of filaments, but not

$\mathbb{B}$ from $\mathbb{C}$; we can distinguish $\mathbb{B}$ from $\mathbb{C}$ based on whether there are split anthers (the ends), but not $\mathbb{C}$ from $\mathbb{A}$, and so on. To recognise classes it is necessary to consider their attributes in the context of other attributes.

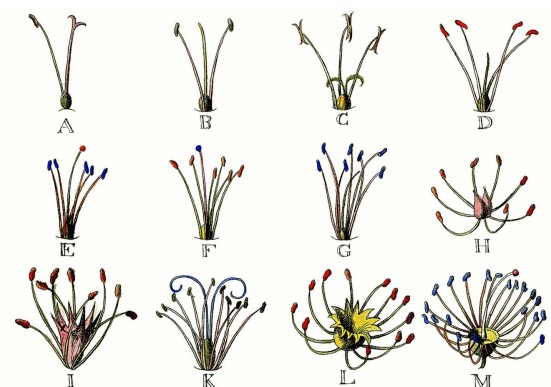

Figure 1: Section of *Methodus Plantarum Sexualis*, Georg Ehret, illustrating Linnaeus's *Systema Naturae*, 1735. Class defining attributes need not be exclusive or universal, and useful classifications may be contextual.

Threshold functions are a related concept in Boolean algebra. A threshold function evaluates positively if a weighted sum of binary inputs crosses some threshold. Lacking a preferred feature basis, we can identify monothetic classifications with threshold functions, as threshold functions that are polythetic in one basis, such as logical $\text{OR}(x, y)$, may be recast in a basis where they are monothetic e.g. binary $\text{OR}(x, y)$ evaluates as the unary $\text{MIN}(x + y, 1)$. We can compare the frequency of monothetic and polythetic classifications in this general case. The number of binary inputs of length $n$ is $2^n$ and the total number of Boolean functions is the number of binary labellings of these inputs, $2^{2^n}$, whereas the number of threshold functions grows only singly exponentially ($\leq 2^{n^2}$), and therefore monothetic classifications represent a vanishingly small proportion of the total (see Gruzling (2007); Irmatov (1993)).

This work explores meta-learning for polythetic classification. Specifically, we

- consider the limitations of widely used threshold classifiers, such as Prototypical Networks (Snell et al., 2017), and how they are able to learn and approximate non-threshold functions in practice;
- show that simple alternatives based on attention, such as Matching Networks (Vinyals et al., 2016), are polythetic by default but susceptible to misclassification due to excessive sensitivity;
- characterise the challenge of spurious correlations in irrelevant features for attentional classifiers;
- and propose a simple solution to this challenge that is non-parametric and based on self-attention.

Throughout, we evidence these findings and the effectiveness of our proposals with experiments on synthetic and real-world few-shot learning tasks.

## 2 BACKGROUND

**Problem formulation.** We are interested in few-shot classification: provided with a small number of labelled points $\mathbb{S} = \{\boldsymbol{x}_i, y_i\}_{i \in \mathbb{I}_\mathbb{S}}$, the support set, with feature vectors $\boldsymbol{x}_i \in \mathbb{R}^n$ and labels $y_i \in \{1, \ldots, K\}$, we want to predict the labels of the query set $\mathbb{Q} = \{\boldsymbol{x}_j\}_{j \in \mathbb{I}_\mathbb{Q}}$. $\mathbb{S}_k$ denotes the set of support elements with label $k$ and $\mathbb{I}_\star$ is the index set of the subscript e.g. $\mathbb{I}_\mathbb{S}$ is the index set of the support. The label space is arbitrary and potentially unique to a task (also referred to as an episode) — both the number of classes, $k$, and assigned labels may vary over tasks — and, importantly, examples that share labels under one categorisation will not necessarily share labels under another. We will refer to classification functions over $n$ features simply as classifications, and in meta-learning we are often interested in classifications that only depend on some features, $\alpha$, and not the remainder, $\beta = n - \alpha$.

**Classifiers.** Deep neural models for problems of this kind are usually equipped with either a threshold classifier or an attentional classifier. Threshold classifiers are based, as the name suggests, on using thresholds to partition a space into regions associated with each class. Prototypical Networks (Snell et al., 2017) are an example of such a model. This model embeds examples using a learned neural function, $f_\phi$, finds the average embedding for each class in the support, $\boldsymbol{c}_k$, and classifies queries based on their proximity (measured with a

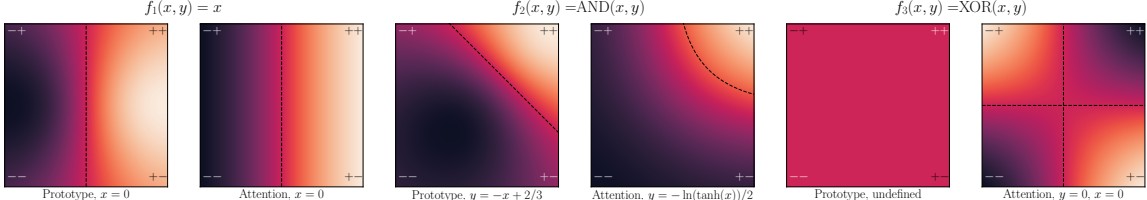

Figure 2: Confidence heatmaps and decision boundaries of prototype and attention classifiers on 2-variable Boolean functions. The attention classifier shown uses a temperature of 1, lower temperatures 'harden' the classification and produce decision boundaries more closely aligned with the axes (see Appendix B). As $\text{XOR}(x, y)$ is not a threshold function, simple prototypes fail to produce a correct classification scheme, in this case the prototypes are equal ($= (0, 0)$) and there is no decision boundary.

distance function $d$) to these class prototypes:

$$
\boldsymbol{c}_k = \frac{1}{|\mathbb{S}_k|} \sum_{i \in \mathbb{I}_{\mathbb{S}_k}} f_\phi(\boldsymbol{x}_i) \quad ; \quad p_\phi(y = k | \boldsymbol{x}) = \frac{\exp(-d(f_\phi(\boldsymbol{x}), \boldsymbol{c}_k))}{\sum_{k'} \exp(-d(f_\phi(\boldsymbol{x}), \boldsymbol{c}_{k'}))} \tag{1}
$$

The key advantage of such an approach is that salient class features are preserved when forming the prototype while irrelevant aspects of particular examples are washed out. Attentional classifiers instead use a similarity function to directly compare queries with each example in the support. For example, Matching Networks (Vinyals et al., 2016) also learn embedding functions, but each query is compared with every member of the support to weight a sum over their labels, which we can write simply as $\hat{y} = \sum_{i \in I_S} a(\hat{x}, x_i) y_i$. In the popular terminology of transformers we may write the embedded queries as $\mathbf{Q}$, the embedded support as keys $\mathbf{K}$ and their labels as values $\mathbf{V}$, and dot-product attention classification with temperature $\tau^{-1}$ as

$$
\text{DotAttn}(\mathbf{Q}, \mathbf{K}, \mathbf{V}, \tau) = \text{softmax}(\tau \mathbf{Q} \mathbf{K}^T) \mathbf{V} \in \mathbb{R}^{|\mathbb{Q}| \times k}. \tag{2}
$$

Attentional classifiers are more sensitive to variations within a class at the cost of additional computation.

**Boolean tasks.** In comparing these classifiers we make repeated use of tasks based on Boolean functions, and on exclusive-OR (XOR) in particular. For a binary feature vector $\boldsymbol{x} \in \{-1, 1\}^n$, the number $\chi_\mathbb{A}(\boldsymbol{x}) = \prod_{i \in \mathbb{A}} x_i$ is the parity function or exclusive-or (XOR) over the bits $(x_i)_{i \in \mathbb{A}}$. We write a parity function of $\alpha$ bits $\text{XOR}_\alpha$. The set of parity functions over $n$ bits form a linearly independent basis (O'Donnell, 2014) and, as such, being able to model the partition functions guarantees that one can model any other Boolean function over that domain. Put another way, the decision boundaries of $\text{XOR}_\alpha$ are at least as complicated as those for any other $\alpha$-variable Boolean function: there is one between every possible pair of feature vectors. For these reasons, XOR is our polythetic function of choice in derivations, examples, and experiments.

## 3 CHALLENGES IN META-LEARNING POLYTHETIC CLASSIFICATIONS

Threshold and attentional classifiers have their own strengths and, in meta-learning polythetic classifications, their own weaknesses. Threshold classifiers are insufficiently flexible and attentional classifiers are prone to misclassification.

**Threshold classifiers.** Figure 2 shows the decision boundaries formed by a prototypical threshold classifier and an attentional classifier for a selection of 2-variable Boolean functions, highlighting the problem with using threshold classifiers for polythetic classification: logical $\text{XOR}(x, y)$ is not a threshold function and the prototypes fail to produce a useful decision boundary. This is the perceptron problem identified by Minsky &

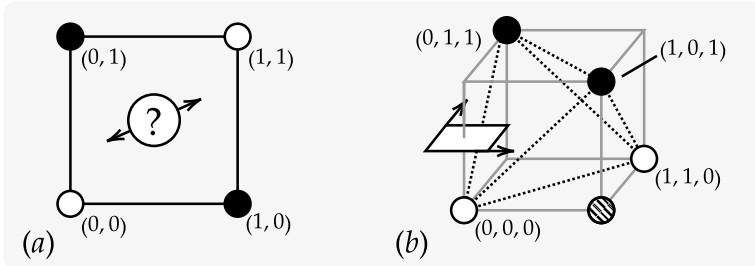
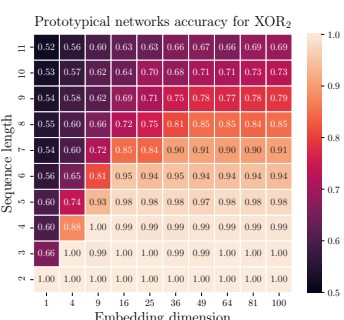

Figure 3: A non-threshold function of 2-variables, the pseudo-variable solution, and Prototypical Network performance on the $XOR_2$ problem. **(a)** $XOR(x, y)$, which does not have a threshold solution in 2 dimensions. **(b)** Appending the pseudo-variable $XOR(x, y)$ gives a 3-dimensional embedding in which all 2-variable Boolean functions have threshold solutions. $XOR(x, y)$ is a pseudo-variable in that it is determined by the other variables and cannot freely vary, for example the hatched circle at $(1, 0, 0)$ cannot occur. *Right:* Accuracy of prototypical networks for the $XOR_2$ problem over sequence length and embedding dimension. Mean over 1000 tasks, $|\mathbb{S}| = 40$. See Appendix F for details.

Papert (1969). However, deep networks using threshold classifiers can learn XOR. This is possible because the network can learn additional pseudo-features for the non-threshold functions it observes. Figure 3 shows an example for 2-variables: one can embed the corners of the square at the corners of a tetrahedron with coordinates $(x, y, XOR(x, y))$ and produce linear thresholds solutions for every 2-variable Boolean function.

There are two problems with this approach, both of which are exacerbated in the meta-learning context: i) the required number of pseudo-variables grows as $\binom{n}{\alpha} \sim \mathcal{O}(n^\alpha)$ to account for all combinations of $\alpha$ active components, and ii) the method does not generalise to unseen non-threshold functions (see Appendix A). The right plot in Figure 3 demonstrates these shortcomings for $XOR_2$: the required number of pseudo-variables is $\binom{n}{2} \sim \mathcal{O}(n^2)$ and initially we find that a quadratic embedding is able to maintain performance, but for longer sequences the number of threshold-functions unseen in training grows and performance degrades.

**Attentional classifiers.** Attentional classifiers avoid these problems — the required embedding dimension is linear in the number of features and the classifier generalises to unseen classifications — but suffer from over-sensitivity to irrelevant features. This results in misclassification, which we quantify in the case of Boolean functions to understand the scaling properties of this problem generally.

Consider classifications over binary feature vectors $\boldsymbol{x} \in \{-1, 1\}^n$, with the class determined by $XOR_\alpha$ over $\alpha$ elements with the remaining $\beta = n - \alpha$ being irrelevant. Assume each of the $2^\alpha$ variations of the active elements are present with equal frequency, $r$, for a support set $\mathbb{S}$ of size $|\mathbb{S}| = r2^\alpha$, and that the remaining $\beta$ elements follow a Bernoulli distribution with probability $p$. Using an attention classifier of the form $\hat{y} = \sum_{i \in \mathbb{S}} \text{softmax}_i \left( a(\hat{x}, x_i) \right) y_i$, where $a(\hat{x}, x_i)$ is a measure of the similarity of $\hat{x}$ and $x_i$, how likely is it that we misclassify a query drawn from the same distribution as $\mathbb{S}$?

Without loss of generality, we can focus on the positive examples (with label $= 1$) for which a positive output gives the correct classification. Using dot-product attention, the mean and variance of the classifier output are, with $\bar{p} = p^2 + (1 - p)^2$ and $\bar{q} = 1 - \bar{p}$,

$$\mu = r(e - e^{-1})^\alpha \left[ \left( \bar{p}e + \bar{q}e^{-1} \right)^\beta \right] = r(e - e^{-1})^\alpha [c^\beta], \tag{3}$$

$$\sigma^2 = r(e^2 + e^{-2})^\alpha \left[ \left( \bar{p}e^2 + \bar{q}e^{-2} \right)^\beta - \left( \bar{p}e + \bar{q}e^{-1} \right)^{2\beta} \right] = r(e^2 + e^{-2})^\alpha [d^\beta - c^{2\beta}], \tag{4}$$

introducing $c$ and $d$ for compactness. The mean is positive, as desired, but we are interested in the rate of misclassification, which may be interpreted using the scale-free and dimensionless coefficient-of-variation (the ratio of the standard deviation to the mean) where greater variation indicates a greater rate of misclassification. From Equations 3 and 4, we have

$$\frac{\sigma}{\mu} = \frac{1}{\sqrt{r}} \left( \frac{\sqrt{e^2 + e^{-2}}}{e - e^{-1}} \right)^\alpha \left( \left( \frac{d}{c^2} \right)^\beta - 1 \right)^{1/2}. \tag{5}$$

Starting with the leftmost term, increasing the number of repetitions, $r$, reduces the relative variability. This aligns with intuition and limits, where an empty support set, $r = 0$, provides no basis on which to make predictions and at the other extreme, $r \gg 2^\beta$, the support set is likely to span the input domain reducing classification to look-up. The term raised to $\alpha$ is approximately 3.2, and so the variability increases with the number of active elements. An intuitive explanation is that the number of immediate neighbours of each point grows as $\binom{n}{\alpha} = \alpha$ and this reduces the confidence with which the point is classified, so the barrier to misclassification is reduced. Finally, $d > c^2$ and so the rightmost term is positive and grows exponentially in $\beta$, meaning that misclassification increases with the number of irrelevant features. A full derivation, including alternative attention functions, is provided in Appendix C.

The problem of misclassification due to over-sensitivity in attentional classifiers was recognised in the work of Luong et al. (2015) on sequence processing. There the problem was addressed by attending only to a subset of elements within some distance of the target position. However, sets do not have such an ordering and so we instead propose a feature-selection method to resolve the problem more generally.

## 4 ATTENTIONAL FEATURE SELECTION

A key challenge of meta-learning is that not all features are relevant in all tasks and that the support is unlikely to span the input domain. The model must choose, using incomplete information, what to focus on and what to ignore by detecting the salient features within and between classes. For monothetic classifications this is straightforward: by definition, averaging highlights necessary features whilst diminishing irrelevant features. Prototype methods rely on this process. In the polythetic case, attentional classifiers have the advantage of being able to learn non-threshold functions without the need for pseudo-variables but do not benefit, as prototypes do, from 'washing-out' irrelevant features through averaging. Indeed, attentional classifiers are susceptible even in the monothetic setting to misclassifying on the basis of closely matching features that

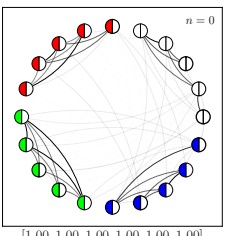 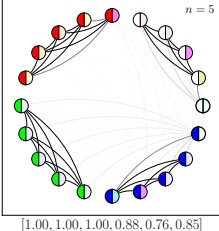 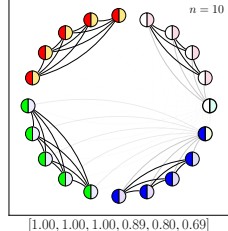 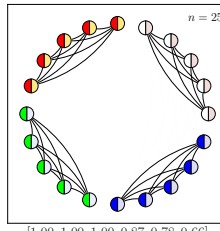

Figure 4: Feature values and attention coefficients during the feature-selection self-attention ($n = \{0, 5, 10, 25\}$) within a class of XOR$_3$. Nodes depict examples of the support set: the colour of the left halves represents the active features; the right halves represents the magnitude of the irrelevant features. Edge width and opacity indicate the attention strength between a pair of nodes. The red, green, blue and white groups, different variants, automatically segregate which preserves their active features while the irrelevant features converge, as shown in the feature scores beneath each plot. In this way, the active features identify themselves.

are not relevant to the problem (putting a zebra with the big-cats because its stripes match those of a tiger in the support set, for example). Misclassification occurs when irrelevant features overwhelm the signal from the active elements. As this is a problem of highlighting the salient patterns within a set, we propose a self-attention based mechanism for feature selection, presented in Algorithm 1 and illustrated in Figure 4.

Intuitively, the process exploits the over-representation of patterns within features that are relevant to the classification as compared to patterns within the irrelevant features. We first standardise the features to prevent those common to the entire support, which are not discriminating, from dominating (Line 1) and stabilise, $\epsilon$, to prevent weakly activated features from being excessively scaled-up. We then repeatedly self-attend within each separate class $k$ of the support set, using dot product attention with scale $\tau$, $\mathbf{X}_k \leftarrow \text{DotAttn}(\mathbf{X}_k, \mathbf{X}_k, \mathbf{X}_k, \tau)$. Self-attention maps elements of a set of vectors to the interior of their convex hull. If every member of a class has some feature in common, the convex hull in that dimension is a point and the features do not change. In the polythetic case it is patterns of features that matter, and by attending more strongly between elements of the support set with such feature-patterns, these too are preserved. Figure 4, for example, shows polythetic variations within a class of $\text{XOR}_3$ with three active and three irrelevant features ($\alpha = \beta = 3$). The patterns in the active features self-reinforce, forming cliques of strongly connected elements, whilst the irrelevant features decay. Finally, features are scored by their dispersion over the support set (Line 5) which indicates how well they have been preserved through the self-attention iterations, and thus how relevant they are.

---

**Algorithm 1:** Self-attention feature scoring. Scores can be used for rescaling or masking. Note that the z-normalisation is over the entire support set whilst the self-attention is within classes. The choice of dispersion measure is of secondary importance and discussed in the main text.

---

**Input** : Support set $S = \{\mathbf{x}_i, y_i\}_{i \in I_S}$ with class labels $y_i \in \{1, \dots, K\}$ and features $\mathbf{x}_i \in \mathbb{R}^F$, $S_k$ denoting the subset of $S$ containing all samples with $y_i = k$ and $\mathbf{X}_k \in \mathbb{R}^{|S_k| \times F}$ an arbitrarily ordered matrix of feature vectors belonging to $S_k$; small numerical constant, $\epsilon$; attention temperature, $\tau^{-1}$; repetitions, $R$.

**Output** : Feature scores, $\boldsymbol{f} \in \mathbb{R}^F$.

1 $\mathbf{x}_i \leftarrow (\mathbf{x}_i - \mu_X)/(\sigma_X + \epsilon)$ ;            // standardise
2 **repeat R times**
3     **for** $k \leftarrow 1$ **to** $K$ **do**                 // for each class
4        $\mathbf{X}_k \leftarrow \text{softmax}(\tau \mathbf{X}_k \mathbf{X}_k^T)\mathbf{X}_k$ ;        // softmax is row-wise
5 $\boldsymbol{f} \leftarrow \text{dispersion}(\{\mathbf{x}_i\}_{i \in S_i})$ ;      // mean-absolute-deviation, std. dev etc.

---

The scores can be used directly to rescale features across the support and query sets before applying the classifier, as in Figure 5, or in top-k selection. We focus on rescaling as the method that makes the fewest assumptions about the underlying classification, but using top-k is highly effective when the number of active elements is known, as shown in Appendix D.

## 5 EXPERIMENTS

We compare the proposed method (FS) with Prototypical Networks (PN) (Snell et al., 2017), a threshold classifier, and Matching Networks (MN) (Vinyals et al., 2016), an attentional classifier without feature-selection, in a sequence of increasingly complex synthetic and real-world few-shot learning problems. As our approach is non-parametric and operates directly on high-level features, it is agnostic to the choice of feature extractor, and we are free to choose as appropriate e.g. a convolutional neural network for images or a multi-layer perceptron for tabular data, and in all experiments we use the same embedding model for all methods (see experimental details in Appendix F).

**Binary strings.** We consider meta-learning Boolean functions of $n = \alpha + \beta$ variables. Labels for inputs $\boldsymbol{x} \in \{-1, 1\}^n$ are generated by computing the XOR of a random subset of components of size $\alpha$. Each variation in the active elements occurs 5 times in the support, $|\mathbb{S}| = 5 \cdot 2^\alpha$. The subset of active components

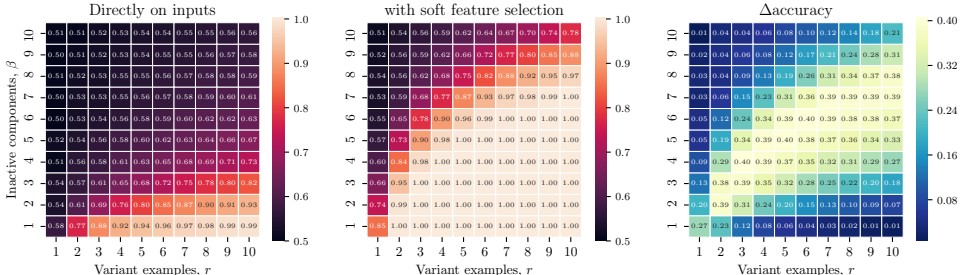

Figure 5: Attention classification of the $XOR_4$ problem over variant frequency, $r$, and number of inactive components $\beta$. Increasing $r$ assists feature selection, in agreement with the derived misclassification distribution. Soft feature-selection rescales features according to their scores, as determined by the proposed self-attention procedure. This greatly improves performance even at low repetitions, for example at 2 repetitions and 3 inactive components the change in accuracy is +38pp. Neither method is effective at high $\beta$ with low $r$.

Table 1: Binary strings. Accuracy by embedding dimension for sequences of length $n = 5$ and $n = 10$. Mean and standard error calculated over 1000 tasks.

| Model | Emb. | $n = 5$ | | | $n = 10$ | | |
|---|---|---|---|---|---|---|---|
| | | $XOR_2$ | $XOR_3$ | $XOR_4$ | $XOR_2$ | $XOR_3$ | $XOR_4$ |
| | 1 | $57.6 \pm 0.5$ | $55.3 \pm 0.5$ | $60.8 \pm 0.7$ | $51.7 \pm 0.4$ | $50.2 \pm 0.2$ | $50.1 \pm 0.2$ |
| PN | $n$ | $73.6 \pm 0.5$ | $70.3 \pm 0.7$ | $91.4 \pm 0.4$ | $56.7 \pm 0.4$ | $50.1 \pm 0.2$ | $50.4 \pm 0.2$ |
| | $n^2$ | $90.4 \pm 0.3$ | $77.8 \pm 0.7$ | $\underline{100.0 \pm 0.0}$ | $62.1 \pm 0.4$ | $50.3 \pm 0.2$ | $50.6 \pm 0.2$ |
| FS+MN | $n$ | $\mathbf{99.6 \pm 0.2}$ | $\mathbf{100.0 \pm 0.0}$ | $\underline{100.0 \pm 0.0}$ | $\mathbf{75.9 \pm 1.3}$ | $\mathbf{82.6 \pm 1.1}$ | $\mathbf{96.3 \pm 0.5}$ |

is unknown to the meta-learner. Appendix I shows a concrete example of an $XOR_2$ task. Table 1 summarises the performances of Prototypical Networks and our approach. PN accuracy decreases sharply with sequence length $n$ and the number of embedding units required to effectively solve this problem grows rapidly with $n$, as shown previously in Figure 3. This suggests that PN are indeed learning pseudo-variables and demonstrates the limitations of threshold classifiers in solving polythetic problems.

**Polythetic MNIST.** We evaluate the ability of the models to jointly extract high-level features and identify polythetic patterns. We build tasks (episodes) using MNIST digits (LeCun et al., 2010), where an example consists of 4 coloured digits (RGB). An example task and further details are provided in Appendix J. For monothetic tasks, a single high-level feature (e.g. colour of the top-right digit) distinguishes classes. For polythetic tasks, class membership derives from XOR interactions over a subset of features, and the remainder are problem-irrelevant. Table 2 shows the performances on three versions of the polythetic MNIST dataset: clean (excluding non-discriminative digits), colourless (task-irrelevant digits but no colour), and full (both task-irrelevant digits and colour). The models are trained on monothetic tasks and evaluated both on monothetic and polythetic tasks. Protonets excel at identifying monothetic features and ignoring non-discriminative features, but have a close to random performance on polythetic tasks. Conversely, matching networks, which are polythetic classifiers by default, are highly sensitive to task-irrelevant features. The proposed approach (FS) can simultaneously detect salient features and perform polythetic classifications. Furthermore, as shown in Figure 6, we found our classifier to be robust to the rate of polythetic tasks seen during training in a second experiment.

**Omniglot.** The Omniglot dataset (Lake et al., 2011) consists of handwritten characters from 50 writing systems with 20 hand drawn examples of each character. Training tasks are formed using examples from 30 of the alphabets and test tasks draw from the other 20. We compare our method to PN, MN, Infinite Mixture

Table 2: Polythetic MNIST. Evaluation accuracy on monothetic and polythetic tasks in three settings. Mean and standard error calculated over 1000 tasks.

| | Clean | | Colourless | | Full | |
|---|---|---|---|---|---|---|
| Model | Monothetic | Polythetic | Monothetic | Polythetic | Monothetic | Polythetic |
| PN | **97.9 ± 0.1** | 50.6 ± 0.3 | 92.8 ± 0.3 | 49.9 ± 0.3 | **94.5 ± 0.3** | 49.8 ± 0.2 |
| MN | 79.6 ± 0.6 | 57.6 ± 0.4 | 69.7 ± 0.4 | 61.0 ± 0.5 | 70.1 ± 0.7 | 56.6 ± 0.5 |
| FS+MN | 96.8 ± 0.1 | **98.3 ± 0.0** | **94.5 ± 0.2** | **98.0 ± 0.0** | 75.0 ± 0.7 | **60.4 ± 0.7** |

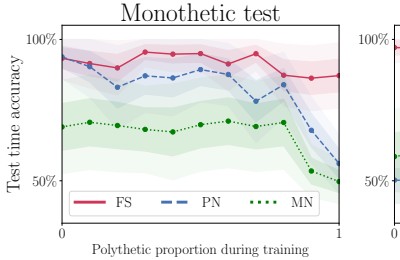

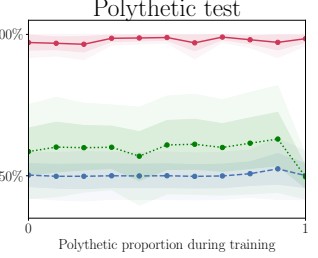

Figure 6: Polythetic MNIST (colourless) by polythetic proportion during training. FS matches or outperforms the other models at all training proportions and is far less affected by the training mix. Mean and standard deviation over 1000 tasks at each proportion.

Table 3: Omniglot. 20-way, 5-shot characters; 3-way alphabets. Mean and standard error on 1000 tasks.

| | Characters | Alphabets |
|---|---|---|
| PN | **98.6 ± 0.0** | 83.4 ± 0.3 |
| MN | 91.1 ± 0.1 | 78.4 ± 0.3 |
| FS+MN | 96.2 ± 0.0 | 94.2 ± 0.2 |
| IMP | **98.6 ± 0.0** | **96.0 ± 0.2** |
| MAML | 94.0 ± 0.1 | 89.9 ± 0.3 |
| MN* | 97.9 ± 0.1 | 81.3 ± 0.3 |
| NN* | 98.3 ± 0.0 | 95.7 ± 0.3 |
| FS+MN* | 98.1 ± 0.0 | **96.0 ± 0.2** |
| FS+NN* | 98.3 ± 0.0 | **96.0 ± 0.2** |

Prototypes (IMP) (Allen et al., 2019), and MAML (Finn et al., 2017) with a threshold classifier (Triantafillou et al., 2020). We train the models for character recognition, and additionally evaluate performance on 3-way alphabet recognition (inherently polythetic). Our approach can be used in conjunction with most few-shot learning approaches – we specifically apply FS prior to MN and single-nearest-neighbours (NN), corresponding to MN with softmax converging to argmax (see Appendix B). Table 3 shows the results of this experiment. The end-to-end trained model (FS+MN) is competitive with PN and IMP, while performing better than MN and MAML in character recognition. In alphabet recognition, FS+MN performs better than other methods, while being competitive with IMP. We further evaluate the performance when using a pre-trained feature extractor (methods marked with *), obtained by training a PN threshold classifier on character recognition. FS+MN* improves over MN* in both tasks. Compared to PN, the accuracy of FS+MN* and FS+NN* in character recognition is reduced by at most 0.5pp; yet improved in alphabet recognition by 12.6pp, while performing similarly to the more complex IMP.

**TieredImageNet.** TieredImageNet (Ren et al., 2018) is a subset of ILSVRC-12 (Russakovsky et al., 2015) with polythetic characteristics, with classes grouped into categories corresponding to higher-level nodes in the ImageNet hierarchy. There are 34 categories of 10 to 30 classes each. We compare our method (FS) to

Table 4: TieredImageNet. Model accuracy by categories (C) and groups (G) over 500 tasks.

| G | | C = 2 | C = 4 | C = 8 |
|---|---|---|---|---|
| 5 | FS | **83.5 ± 0.3** | **66.4 ± 0.3** | 48.9 ± 0.2 |
| | MN | 82.7 ± 0.4 | 65.4 ± 0.3 | 48.3 ± 0.2 |
| | PN | 81.4 ± 0.3 | 64.6 ± 0.3 | **49.4 ± 0.1** |
| 10 | FS | **83.6 ± 0.3** | **65.8 ± 0.3** | **48.7 ± 0.1** |
| | MN | 82.9 ± 0.3 | 64.9 ± 0.3 | 48.0 ± 0.1 |
| | PN | 81.1 ± 0.3 | 63.3 ± 0.3 | 48.2 ± 0.1 |

Table 5: TieredImageNet. Head-to-head comparison over 500 tasks by categories (C) and subgroups (G). **Bold** indicates significance at the $p < 0.001$ level.

| | | | C = 2 | | C = 4 | | C = 8 | |
|---|---|---|---|---|---|---|---|---|
| G | X | Y | X / Y | (tie) | X / Y | (tie) | X / Y | (tie) |
| 5 | FS | MN | **230** / 162 | (108) | **329** / 101 | (70) | **343** / 113 | (44) |
| | FS | PN | **319** / 136 | (45) | **339** / 142 | (19) | 239 / 247 | (14) |
| 10 | FS | MN | **258** / 185 | (57) | **357** / 108 | (35) | **413** / 68 | (19) |
| | FS | PN | **374** / 101 | (25) | **396** / 99 | (5) | **296** / 191 | (13) |

PN and MN classifiers. We use a publicly available pre-trained ResNet-12 (Zhang et al., 2020), pre-trained using the training classes in TieredImageNet, as the feature extractor for all models. Table 4 presents the aggregate accuracy while Table 5 shows the head-to-head results of this experiment. FS leads to significant improvements in performance (except C=8/G=5, where the difference between PN and FS is not significant) in this full scale, naturally polythetic problem, particularly in the "more polythetic" case with 10 subgroups.

## 6 RELATED WORK

We characterise meta-learning approaches for few-shot classification. In addition to evaluating their ability to generalise to unseen classes, we investigate how well they can adapt to polythetic tasks (generalisation to unseen ways of cateogorising). Adaptability in few-shot settings has been studied through different paradigms such as fast weights (Ba et al., 2016), learnable plasticity (Miconi et al., 2018) and meta-learning. Recent work on meta-learning for few-shot classification includes approaches that are able to quickly adapt through various mechanisms such as recurrent architectures (Mishra et al., 2018; Santoro et al., 2016) for learning parameter updates (Ravi & Larochelle, 2017). Other more general optimisation-based approaches (Finn et al., 2017; Nichol et al., 2018; Rusu et al., 2019), tackle these tasks by explicitly optimising the model's parameters. These, however, are typically model-agnostic and commonly used in conjunction with threshold classifiers (Triantafillou et al., 2020), inheriting their limitations in a polythetic scenarios.

Our work aligns more closely with metric-learning approaches for few-shot classification (Chen et al., 2019) that apply distance functions between queries and the support in a common embedding space (Allen et al., 2019; Oreshkin et al., 2018; Snell et al., 2017; Sung et al., 2018; Vinyals et al., 2016; Zhang et al., 2020). Methods that construct class-wise prototypes from the support (Allen et al., 2019; Ren et al., 2018; Snell et al., 2017) can successfully tackle monothetic tasks, but can struggle with task-adaptiveness in a polythetic context. Attentional meta-classifiers (Hou et al., 2019; Jiang et al., 2020; Kim et al., 2019; Vinyals et al., 2016) adapt to polythetic tasks but lack crucial mechanisms for focusing exclusively on relevant features.

Attending over datapoints has been considered previously. Luong et al. (2015) introduced dot-product attention in the context of attending over sequences. Vinyals et al. (2016) considered $\hat{y} = \sum_{i \in I_S} a(\hat{x}, x_i) y_i$ and provided the conditions under which such a model carries out kernel density estimation or $k$-nearest-neighbours classification. Vaswani et al. (2017) used explicit scaling in hidden attention layers, but scaling a classifying softmax by a temperature parameter dates to the work of Boltzmann and later Gibbs (1902). Plötz & Roth (2018) note that their neural-nearest-neighbours ($N^3$) block recovers a soft-attention weighting when the number of neighbours is set to 1, and deploy their model on an outlier-detection set-reasoning task.

## 7 CONCLUSION

In this work we have articulated the difference between monothetic and polythetic classifications and considered the limitations of standard meta-learning classifiers in the polythetic case. We have shown that threshold classifiers require an embedding space that is exponential in the number of active features and that attentional classifiers are overly sensitive and susceptible to misclassification. To address this, we have proposed an attention based method for feature-selection and demonstrated the effectiveness of our approach in several synthetic and real-world few-shot learning problems. Our approach is simple and can be used in conjunction with most few-shot meta-learners. We expect polythetic meta-learners to find real-world application in domains where data is typically scarce and complex, such as healthcare or bioinformatics. For example, we envision a use in classifying rare diseases — there are around 7000 rare diseases, affecting $\sim$1/17 of the worldwide population — from DNA sequences, where mutations often lead to different phenotypes. In such scenarios, few-shot learning approaches able to generalise over unseen combinations of mutations (i.e. ways of categorising), in a similar vein to our binary strings experiment, may lead to better performance in diagnosing rare diseases and shed new insights into their molecular mechanisms.

## REPRODUCIBILITY STATEMENT

In Section 3 we discuss the challenges of polythetic classification and the limitations of current work on meta-classifiers. Our proposed self-attention feature scoring algorithm is described in detail in Section 4, and in particular, Algorithm 1 (and Appendix H). To ensure the work is readily reproducible, besides descriptions of the experimental setup provided in Section 5 and the supplementary material (Appendices F, G, I, J) - we also provide code used for producing the results in the paper.

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

## A  PARTITIONED XOR PERFORMANCES

We hypothesise that, in order to solve polythetic tasks, prototypical networks need to create pseudo-variables in the embedding space (e.g. XOR for each pair of components for binary strings tasks). To test this, we train a prototypical network on binary strings tasks (where pairs of active components are always chosen from the 5 first components) and evaluate the performance on unseen combinations of components (i.e. combinations of the 5 first components not seen during training). Table 6 shows the accuracies for sequences of length $n = 5 + \beta$ (where $\beta$ is the number of variables that are always inactive). We attribute the better-than-chance performances (i.e. for 100-dimensional embeddings) to the high-dimensionality of the embeddings – by chance, large numbers of non-linear features (e.g. output by a random feature extractor) will include some features that are useful for the task of interest (i.e. similar to extreme learning machines). Overall, these results highlight the inability of prototypical networks to generalise to unseen combinations and support our hypothesis.

Table 6: Accuracy of prototypical networks on unseen non-threshold functions by embedding dimension. We use sequences of length $n = 5 + \beta$, where $\beta$ is the number of components that are always inactive. The labels are derived from $\text{XOR}_2$ combinations over the first 5 components and the sets of combinations seen at train and test times are disjoint. In other words, the combinations of variables $(0, 1), (0, 2), (1, 3), (2, 4), (3, 4)$ are only seen active at train time, whereas, the combinations $(0, 3), (0, 4), (1, 2), (1, 4), (2, 3)$ are only seen at test time, with $r = 5$ repetitions for each XOR combination. Each of the 5 first variables has the same expected frequency of being active at train and test times. The inability to generalise in this scenario suggests that protonets need pseudo-variables (i.e. XOR functions applied to each pair of components) to solve the binary strings task.

| Emb. | $\beta = 0$ | $\beta = 1$ | $\beta = 2$ |
|---|---|---|---|
| 1 | $48.741 \pm 0.276$ | $48.855 \pm 0.282$ | $49.558 \pm 0.284$ |
| 10 | $50.350 \pm 0.282$ | $49.397 \pm 0.286$ | $49.479 \pm 0.265$ |
| 20 | $50.705 \pm 0.271$ | $51.057 \pm 0.247$ | $50.144 \pm 0.255$ |
| 100 | $52.246 \pm 0.256$ | $52.390 \pm 0.265$ | $52.573 \pm 0.258$ |

## B  TEMPERATURE IN ATTENTION CLASSIFICATION

The softmax in attention mechanisms permits a temperature scaling that interpolates between argmax and uniform (and argmin.) This is controlled by $T = \frac{1}{\beta}$, as

$$\text{softmax}_i(\boldsymbol{x}, \beta) = \frac{\exp(\beta x_i)}{\sum_j \exp(\beta x_j)}$$

with softmax converging to argmax as $\beta \to \infty$, and to uniform (i.e. all elements equal to the reciprocal of the length of the vector) as $\beta \to 0$.

For attention classifiers, a decrease in temperature increases the model confidence and can cause decision boundaries to move. As $\beta \to \infty$ and the softmax converges to argmax, the classifier tends to the single nearest neighbour classification scheme; for $\beta = 0$ the classifier returns the support set class balance. For Boolean functions, changes in temperature effect the degree to which decision boundaries are axis-aligned. For example, Figure 2 shows the decision boundary for $f(x, y) = \text{AND}(x, y)$ using a softmax temperature of

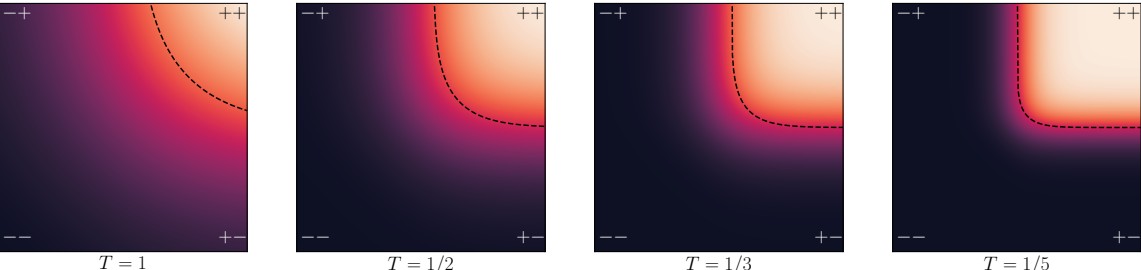

Figure 7: Changes in confidence and decision boundary of the attention classifier with temperature for $\text{AND}(x, y)$.

1 at $y = -\frac{1}{2} \ln (\tanh x)$, which we derive as

$$p(\text{class } 1) = p(\text{class } 0)$$

$$\frac{\exp\big(\beta(x + y)\big)}{\sum \exp(...)} = \frac{\exp\big(\beta(-x - y)\big) + \exp\big(\beta(-x + y)\big) + \exp\big(\beta(x - y)\big)}{\sum \exp(...)}$$

$$\exp\big(\beta(x + y)\big) - \exp\big(-\beta(x + y)\big) = \exp\big(\beta(-x + y)\big) + \exp\big(\beta(x - y)\big)$$

$$\sinh(\beta(x + y)) = \cosh(\beta(x - y))$$

$$\sinh(\beta x)\big(\cosh(\beta y) + \sinh(\beta y)\big) = \cosh(\beta x)\big(\cosh(\beta y) - \sinh(\beta y)\big)$$

$$\tanh(\beta x) = \frac{\cosh(\beta y) + \sinh(\beta y)}{\cosh(\beta y) - \sinh(\beta y)}$$

$$\tanh(\beta x) = \exp(-2\beta y)$$

$$y = -\frac{1}{2\beta} \ln \big(\tanh (\beta x)\big).$$

The effect of decreasing the temperature on the decision boundary is shown in Figure 7.

## C  MISCLASSIFICATION DISTRIBUTION FULL DERIVATION

We first present a more detailed derivation for the case of dot-product attention. We will make repeated use of the Binomial theorem

$$(x + y)^n = \sum_{k=0}^{n} \binom{n}{k} x^{n-k} y^k = \sum_{k=0}^{n} \binom{n}{k} x^k y^{n-k}. \tag{6}$$

Recall that we consider classifications over binary feature vectors $x \in \{-1, 1\}^n$, with the class determined by $\text{XOR}_\alpha$ over $\alpha$ elements with the remaining $\beta = n - \alpha$ being irrelevant. Assume each of the $2^\alpha$ variations of the active elements are present with equal frequency, $r$, for a support set $\mathbb{S}$ of size $|\mathbb{S}| = r2^\alpha$, and that the remaining $\beta$ elements follow a Bernoulli distribution with probability $p$. There are $\binom{\alpha}{\delta}$ strings over the active elements that differ from a given example in $\delta$ positions. XOR flips the classification with each difference, so if $\delta$ is even the class is the same, if $\delta$ is odd the class is different (parity).

The class is determined by the sign of the sum of contributions at an even distance subtract those at an odd distance. Putting those into a single sum we have

$$\text{class} = \text{sign}\left(\sum_{\delta=0}^{\alpha} r(-1)^{\delta}\binom{\alpha}{\delta}\exp\left(\alpha - 2\delta\right)\right). \tag{7}$$

We can factor out the number of repetitions, $r$, and as it is positive it doesn't change the sign. We can then rearrange to match the Binomial theorem form and recover the form in the main text:

$$\text{class} = \text{sign}\left(\sum_{\delta=0}^{\alpha}\binom{\alpha}{\delta}e^{\alpha-\delta}\left(\frac{-1}{e}\right)^{\delta}\right) = \text{sign}\left(\left(e - e^{-1}\right)^{\alpha}\right) = +. \tag{8}$$

Next we give the Binomial distribution of the irrelevant feature contribution as $2\text{B}(\beta, \bar{p}) - \beta$ with $\bar{p} = p^2 - (1-p)^2 = 2p^2 - 2p + 1$. Something that is important to the behaviour of the mean and variance later, and breaks the usual symmetry of $p$ and $q = (1-p)$, is that $\bar{p} \geq 0.5$ and is quadratic in $p$ defined by the three points $(p, \bar{p}) = \{(0, 1), (0.5, 0.5), (1, 1)\}$. $\bar{q}$ has the usual definition $1 - \bar{p}$, so $\bar{q} \leq 0.5$ and so on. The contribution at a difference $\delta$ is then $\text{X}(\delta) \sim \exp\left(\alpha - 2\delta + 2\text{B}(\beta, \bar{p}) - \beta\right)$. The expectation is computed in the usual way

$$\mathbb{E}[\text{X}(\delta)] = \sum_i \text{P}(\text{X} = x_i)x_i = \sum_{b=0}^{\beta} \text{P}(\text{B}(\beta, \bar{p}) = b)\exp\left(\alpha - 2\delta + 2b - \beta\right)$$

$$= \exp\left(\alpha - 2\delta\right)\sum_{b=0}^{\beta}\binom{\beta}{b}\bar{p}^b\bar{q}^{(\beta-b)}\exp\left(2b - \beta\right)$$

$$= \exp\left(\alpha - 2\delta\right)\sum_{b=0}^{\beta}\binom{\beta}{b}(\bar{p}e)^b(\bar{q}e^{-1})^{\beta-b} = \exp\left(\alpha - 2\delta\right)\left(\bar{p}e + \bar{q}e^{-1}\right)^{\beta}.$$

Finding the variance in the traditional way, $\text{Var}[\text{X}(\delta)] = \mathbb{E}[\text{X}(\delta)^2] - \mathbb{E}[\text{X}(\delta)]^2$, first $\mathbb{E}[\text{X}(\delta)^2]$ following the derivation for $\mathbb{E}[\text{X}(\delta)]$:

$$\mathbb{E}[\text{X}(\delta)^2] = \sum_i \text{P}(\text{X} = x_i)x_i^2 = \sum_{b=0}^{\beta}\text{P}(\text{B}(\beta, \bar{p}) = b)\exp\left(2\alpha - 4\delta + 4b - 2\beta\right) \tag{9}$$

$$= \exp\left(2\alpha - 4\delta\right)\sum_{b=0}^{\beta}\binom{\beta}{b}\bar{p}^b\bar{q}^{(\beta-b)}\exp\left(4b - 2\beta\right) \tag{10}$$

$$= \exp\left(2\alpha - 4\delta\right)\left(\bar{p}e^2 + \bar{q}e^{-2}\right)^{\beta}. \tag{11}$$

From this we can write the variance

$$\text{Var}[\text{X}(\delta)] = \exp\left(2\alpha - 4\delta\right)\left(\left(\bar{p}e^2 + \bar{q}e^{-2}\right)^{\beta} - \left(\bar{p}e + \bar{q}e^{-1}\right)^{2\beta}\right). \tag{12}$$

Next we want to find the expectation of the sum of the contributions at each difference $\delta$. As pointed out there are $\binom{\alpha}{\delta}$ many strings at a difference of $\delta$. We apply $\mathbb{E}[\sum_i \text{X}_i] = \sum_i \mathbb{E}[\text{X}_i]$ and $\text{Var}[\text{X} - \text{Y}] = \text{Var}[\text{X}] + \text{Var}[\text{Y}]$,

and, remembering to change the sign with each increase in $\delta$,

$$\mathbb{E}\left[\sum_{i\in S}(-1)^{\delta_i}\mathbf{X}(\delta_i)\right] = \sum_{\delta=0}^{\alpha} r(-1)^{\delta}\binom{\alpha}{\delta}\mathbb{E}[\mathbf{X}(\delta)] \tag{13}$$

$$= r\left(\bar{p}e^2 + \bar{q}e^{-2}\right)^{\beta}\sum_{\delta=0}^{\alpha}\binom{\alpha}{\delta}e^{(\alpha-\delta)}\left(\frac{-1}{e}\right)^{\delta} \tag{14}$$

$$= r\left(\bar{p}e^2 + \bar{q}e^{-2}\right)^{\beta}\left(e - e^{-1}\right)^{\alpha}, \tag{15}$$

and

$$\mathrm{Var}\left[\sum_{i\in S}(-1)^{\delta_i}\mathbf{X}(\delta_i)\right] = \mathrm{Var}\left[\sum_{i\in S}\mathbf{X}(\delta_i)\right] = \sum_{\delta=0}^{\alpha} r\binom{\alpha}{\delta}\mathrm{Var}[\mathbf{X}(\delta)] \tag{16}$$

$$= r\left(\left(\bar{p}e^2 + \bar{q}e^{-2}\right)^{\beta} - \left(\bar{p}e + \bar{q}e^{-1}\right)^{2\beta}\right)\sum_{\delta=0}^{\alpha}\binom{\alpha}{\delta}\left(e^2\right)^{\alpha-\delta}\left(e^{-2}\right)^{\delta} \tag{17}$$

$$= r\left(\left(\bar{p}e^2 + \bar{q}e^{-2}\right)^{\beta} - \left(\bar{p}e + \bar{q}e^{-1}\right)^{2\beta}\right)(e^2 + e^{-2})^{\alpha}. \tag{18}$$

## C.1 DIFFERENT ATTENTION MECHANISMS

For dot-product and cosine-similarity attention, 'angular' difference mechanisms, we use $(+, -)$ to encode the input variables. This is because we want the score to be greater when the variables are the same and lesser when they are opposed (if we were to use $(1, 0)$ we'd have $0 \times 0 \neq 1 \times 1$ and $0 \times 1 = 0 \times 0$). With these methods we get

$$f_{\mathrm{dot}}(\delta) = \alpha - 2\delta, \tag{19}$$

$$f_{\cos}(\delta) = 1 - \frac{2\delta}{\alpha}. \tag{20}$$

The change for cosine-similarity introduces a factor of $\exp(1/\alpha)$ to the mean and $\exp(2/\alpha)$ to the variance, and the overall picture doesn't change

$$\mathbb{E}\left[\sum_{i\in S}(-1)^{\delta_i}\mathbf{X}_{\cos}(\delta_i)\right] = r\left(\bar{p}e^2 + \bar{q}e^{-2}\right)^{\beta}\left(e^{1/\alpha} - e^{-1/\alpha}\right)^{\alpha} \tag{21}$$

$$\mathrm{Var}\left[\sum_{i\in S}(-1)^{\delta_i}\mathbf{X}_{\cos}(\delta_i)\right] = r\left(\left(\bar{p}e^2 + \bar{q}e^{-2}\right)^{\beta} - \left(\bar{p}e + \bar{q}e^{-1}\right)^{2\beta}\right)\left(e^{2/\alpha} + e^{-2/\alpha}\right)^{\alpha}. \tag{22}$$

For squared Euclidean distance attention (which coincides with the 'Laplace attention' L1 norm in this encoding), if we encode the inputs as $(1, 0)$ we get $f_{L2}(\delta) = -(\sqrt{\delta})^2 = -\delta$, introducing a factor of $\exp(-\alpha)$ to the mean and $\exp(-2\alpha)$ to the variance, which also doesn't change the overall picture.

## D    TOP-K FEATURE SELECTION

Top-k feature selection masks out all but the $k$ highest scoring features. That is $\boldsymbol{x} \leftarrow \boldsymbol{m} \odot \boldsymbol{x}$ with

$$m_i = \begin{cases} 1 & \text{if } \text{rank}(\boldsymbol{s})_i \leq k \\ 0 & \text{otherwise} \end{cases} \qquad (23)$$

As compared to soft feature selection which rescales as $\boldsymbol{x} \leftarrow \boldsymbol{s} \odot \boldsymbol{x}$, or $\boldsymbol{x} \leftarrow \boldsymbol{s}' \odot \boldsymbol{x}$ with normalised $\boldsymbol{s}'$.

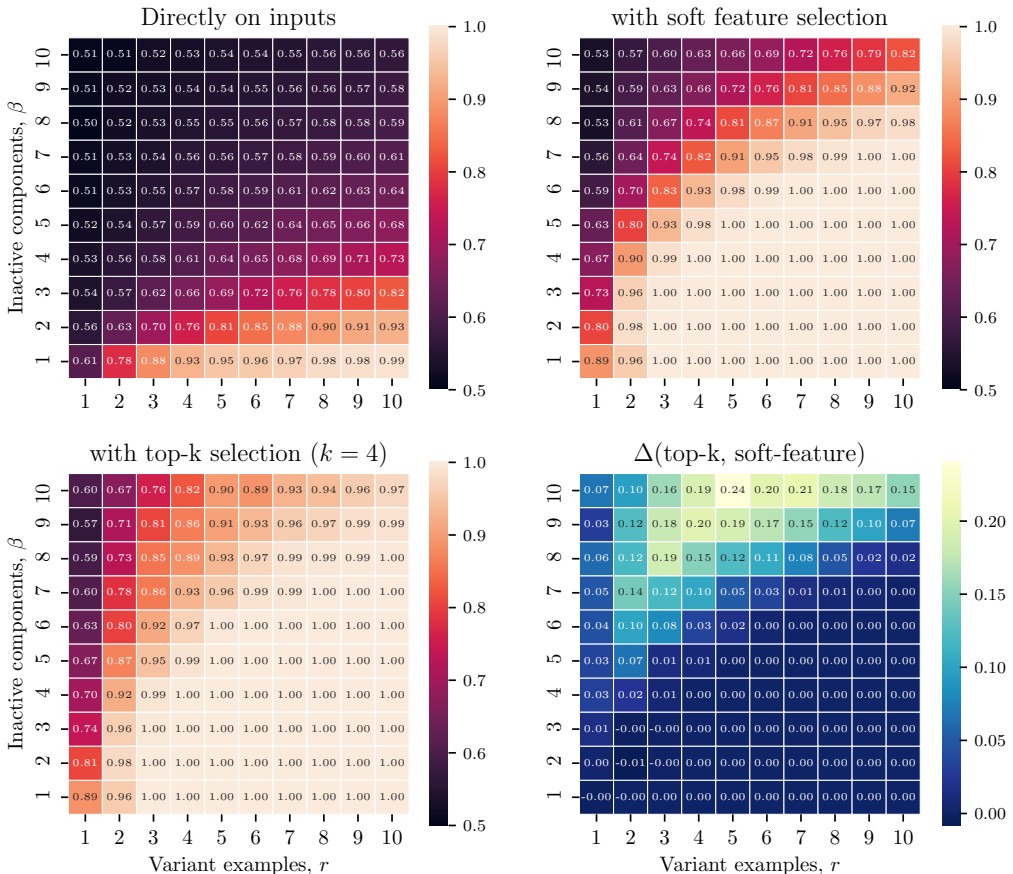

Figure 8: Comparing soft and top-k feature selection in the same setting as Figure 5: classification of $\text{XOR}_4$ over variant frequency, $r$, and number of inactive components $\beta$. Examples of a variant satisfy the XOR in the same way, i.e. the active components are equal, but may not have the same inactive components. The top-k version uses a binary mask to leave the $k$ highest scoring features unchanged and zeroing the rest. This produces significant improvements over even the soft feature selection method at high values of $\beta$, but requires knowledge of the number of active elements.

# E    CONVERGENCE OF FEATURE VECTORS ACROSS SELF-ATTENTION ITERATIONS

Figure 9 illustrates how feature vectors converge on the XOR$_2$ task across self-attention iterations of the feature selection mechanism.

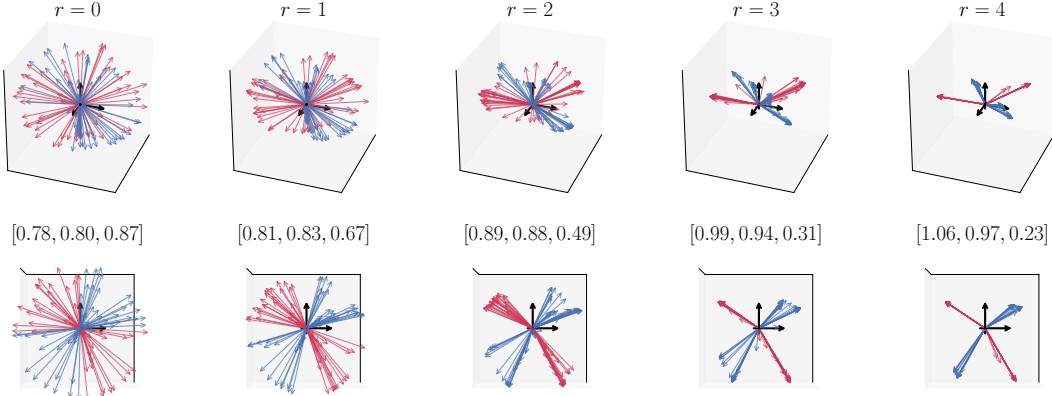

Figure 9: Feature vectors converging under iterated self-attention on a XOR$_2$ classification of vectors uniformly sampled over the sphere, in 3D (top) and down the $z$-axis (bottom). Colours indicate classes. The vectors quickly align by $xy$-quadrant and the variation in $z$ is 'washed-out', also seen in the feature selection scores (mean-absolute-deviation) $[x, y, z]$.

## F  EXPERIMENTAL DETAILS

We use the same feature extractor architecture and train loop for all the baselines.

**Feature extractor.**  We leverage a convolutional neural network with 4 blocks as a feature extractor. Each block consists of a convolutional layer (64 output channels and $3 \times 3$ filters), followed by batch normalisation (momentum 0.01), a ReLU activation, and $2 \times 2$ max pooling:

$$\text{Conv2d}(64, 3 \times 3) \to \text{BN} \to \text{ReLU} \to \text{MaxPool}(2 \times 2)$$

Then, we flatten the output and apply a linear layer to map the data into a 64-dimensional embedding space (unless otherwise stated). As explained in the main manuscript, each method then manipulates these embeddings in different way.

**Train loop.**  We train all models in an episodic manner. At each training iteration, we follow these steps:

- First, we sample task-specific support and query sets. For polythetic MNIST, the support set has 96 samples (2 classes, 2 groups per class, and 24 group-specific examples per group). The query set consists of 32 samples (2 classes, 2 groups per class, and 8 group-specific examples per group). For Omniglot, the support set consists of 5 examples for 20 classes (20-way, 5 shot). The query set consists of 15 examples per class.

- Second, we compute embeddings using the feature extractor and produce class probabilities for the query points. The way in which these probabilities are computed depends on the method (e.g. attentional classification for matching networks or softmax over prototype distances for prototypical networks).

- Finally, we compute the cross entropy for the query examples and optimise the feature extractor via gradient descent. We employ an Adam optimise with learning rate $0.001$.

We train the models for 10,000 iterations (i.e. tasks) for all experiments, except for full polythetic MNIST (100,000 tasks). We then compute the performances on a held-out dataset and average the results across 1,000 tasks.

## G  MULTI-CATEGORICAL PRE-TRAINING

In this experiment we first train a classifier in the multi-categorical setting for the full polythetic MNIST task. In this case there are no task-irrelevant digits or colours as the label describes all four digits with their colours: there are four 10-way labels for the digits ($\in \mathbb{R}^{4 \times 10}$) and four 3-way labels for the colours ($\in \mathbb{R}^{4 \times 3}$) for a combined multi-hot output vector $\in \mathbb{R}^{52}$. The model architectures match that used in the other experiments, see Appendix F, other than using a variation of the MLP head that takes the flattened output from the convolutional network. The variation has two hidden layers with $512$ units each and ReLU activations, before linear layers with softmaxes for each of the label heads. The model is pre-trained over 800 batches of 16 examples drawn at random from the label combinations. In the multi-categorical pre-training, the model achieved a validation accuracy of $95.6\%$ on the digit labels and $100\%$ on the colour labels over 1600 validation examples.

The pre-trained model was then used with prototypical and attentional classifiers in the polythetic MNIST few-shot classification setting discussed in Section 5 and detailed further in Appendix F. We compared performance using the multi-headed softmax activations the model was pre-trained with and an elementwise sigmoid for both the monothetic and polythetic settings. The results are presented in Table 7, and conform to the trend we see in other experiments: threshold classification has the advantage in monothetic tasks but perform no better than chance for polythetic tasks. Attentional classifiers are weaker in the monothetic setting, but more than make up for this defeict in the polythetic setting.

Table 7: Polythetic MNIST problem with multicategorical pre-training. Mean over 1000 tasks.

| | Softmax | | Sigmoid | |
|---|---|---|---|---|
| Model | Monothetic | Polythetic | Monothetic | Polythetic |
| Proto. | 97.95 | 50.18 | 94.64 | 50.21 |
| Attn. | 93.48 | 93.40 | 88.16 | 86.24 |

## H  AN ILLUSTRATIVE EXAMPLE OF THE METHOD

Figure 10 provides a more comprehensive overview of the proposed method. In this example, we are interested in distinguishing between big-cats and equids (horses, donkeys, and in the case zebra). We first extract features from all samples in the support and query sets. Here, we imagine these features as corresponding to some general 'cat' properties, patterns (such as stripes, dots etc.) and general 'equid' properties. Next (as presented in lines $2 : 4$ in Algorithm 1), we perform repeated self-attention with respect to the separate classes of the support set, which yields updated features for each support sample.

We then aggregate the resulting support features with an appropriate dispersion metric, e.g. mean absolute deviation or standard deviation, (line 5 in Algorithm 1) to obtain a vector of feature scores. These scores quantify the relevance of each feature in a given task. Next, we rescale both the query and (initial) support features, i.e. multiplying by the feature scores, to dilute the task-irrelevant and (potentially) misleading features. In this particular example, this would correspond to diluting the 'patterns' feature, since it is irrelevant when distinguishing between cats and equids. Finally, we produce class probabilities via an attentional classifier.

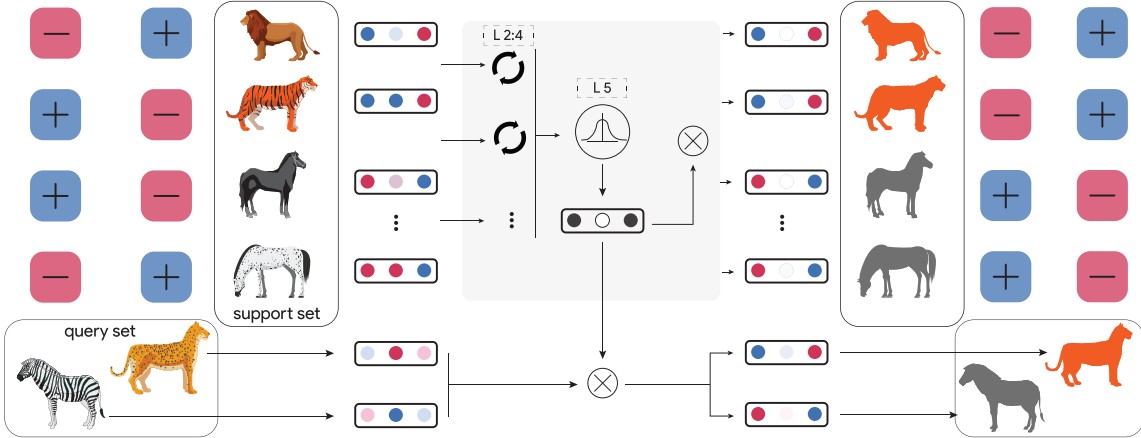

Figure 10: Diagram of the proposed approach. Note how the misleading features of stripes and spots, which may cause misclassification, are diluted through the feature selection process.

## I  CONSTRUCTION OF BINARY STRINGS TASKS

For binary strings tasks, we construct training tasks by sampling $|\mathbb{S}| = 5 \cdot 2^\alpha$ support examples and $|\mathbb{Q}| = 5 \cdot 2^\alpha$ query examples. Each example consists of $\alpha + \beta$ bits. The location of the $\alpha$ active bits is completely random and consistent between the support and query sets within the same task. The labels are computed as the XOR over the $\alpha$ active components. The remaining $\beta$ noisy components are randomly sampled from a Bernoulli distribution with protability 0.5. Figure 11 shows an example of a binary strings task with $\alpha = 2$ active components, $\beta = 3$ noisy bits, and $r = 1$ repetitions.

| | Support set | | | | Query set | |
|---|---|---|---|---|---|---|
| $- - - + -$ | $\rightarrow$ | $-$ | | $+ + - - -$ | $\rightarrow$ | $-$ |
| $+ - - - +$ | $\rightarrow$ | $+$ | | $- + - + +$ | $\rightarrow$ | $+$ |
| $+ + + - -$ | $\rightarrow$ | $+$ | | $- + + + -$ | $\rightarrow$ | $+$ |
| $- + + - +$ | $\rightarrow$ | $-$ | | $- - + + +$ | $\rightarrow$ | $-$ |

Figure 11: Example of an XOR$_\alpha$ task with $\alpha = 2$ active components (3rd and 5th bits), $\beta = 3$ noisy bits, and $r = 1$ repetitions for each combination of active components. The support and query sets contain $r2^\alpha$ examples each.

## J  CONSTRUCTION OF POLYTHETIC MNIST TASKS

For polythetic MNIST tasks, the support set consists of 96 samples, with 48 samples for class 0 and 48 samples for class 1. Each class is further divided into 2 groups of 24 samples and each group is defined by a specific set of traits. The groups are complementary between classes, e.g. red ones and blue zeros for class 0; and blue ones and red zeros for class 1. The set of traits is sampled randomly for each task. The query set is sampled in the same manner, with 2 groups per class and 8 samples per group. Tables 8 and 9 summarise the details of the support and query sets and Figures 13, 14, and 15 show the whole support set for the clean, colourless, and full versions of polythetic MNIST.

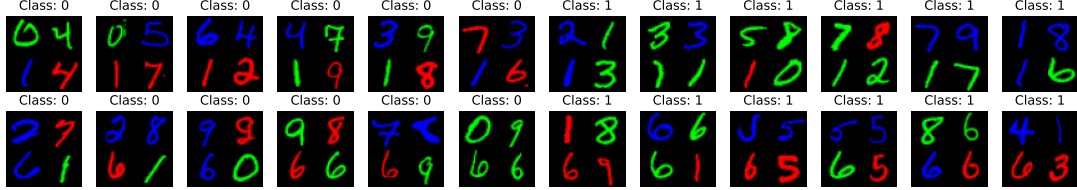

Figure 12: Example of an MNIST polythetic task. Examples from class 0 have either digit 1 (in any colour) in the bottom-left corner with a red digit in the bottom-right corner (top row); or digit 6 in the bottom-left corner with a green digit in the bottom-right (bottom row). Examples from class 1 can have either digit 1 in the bottom-left with a green digit in the bottom-right (top row); or digit 6 in the bottom-left with a red digit in the bottom-right corner (bottom row).

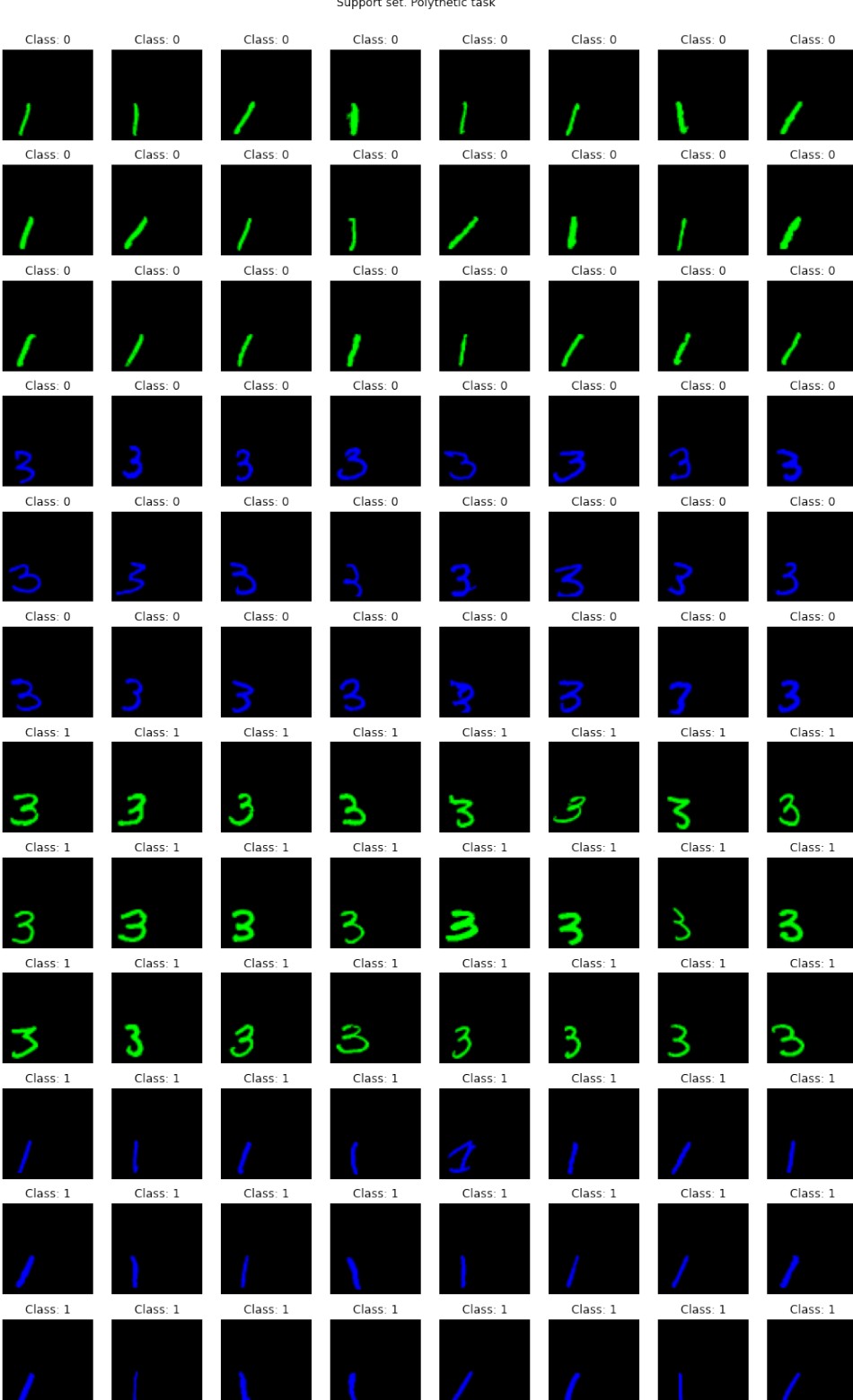

Figure 13: Example of the support set for a polythetic MNIST task (clean).

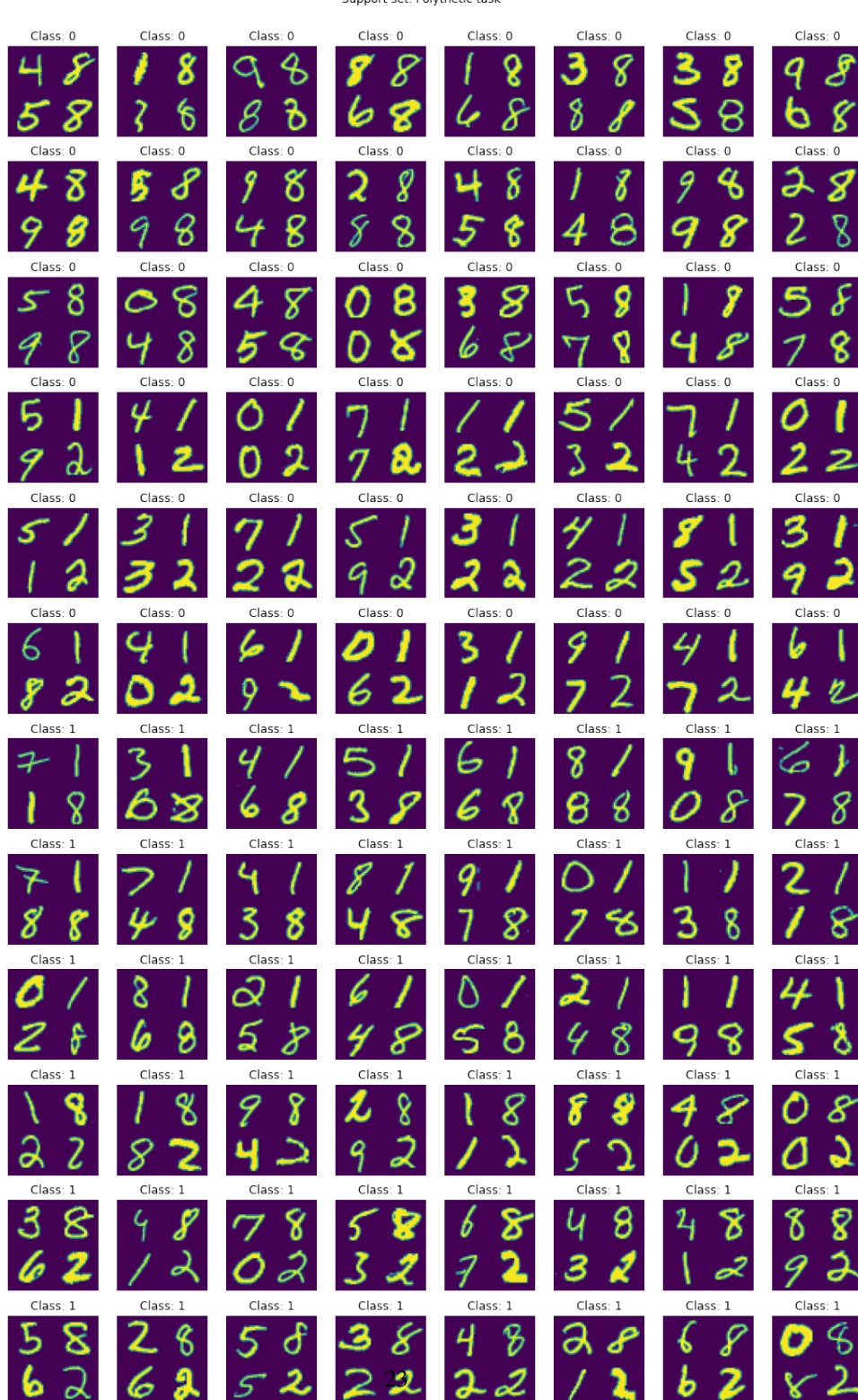

Figure 14: Example of the support set for a polythetic MNIST task (colourless).

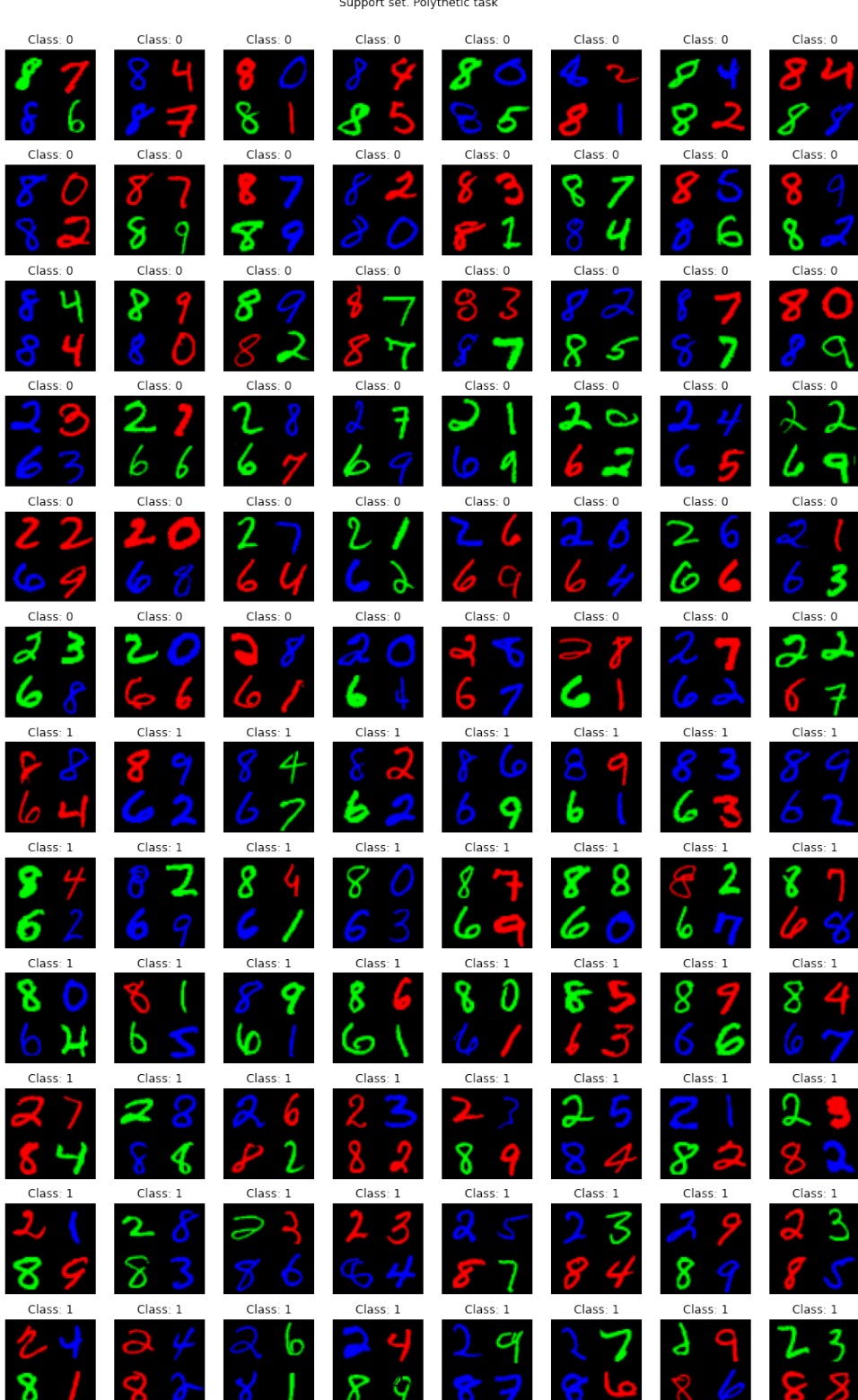

Figure 15: Example of the support set for a polythetic MNIST task (full).

| $|\mathbb{S}| = 96$ | Examples | Groups | Examples per group | Example of groups |
|---|---|---|---|---|
| Class 0 | 48 | 2 | 24 | green ones and blue threes |
| Class 1 | 48 | 2 | 24 | blue ones and green threes |

Table 8: Support set details

| $|\mathbb{Q}| = 32$ | Examples | Groups | Examples per group | Example of groups |
|---|---|---|---|---|
| Class 0 | 16 | 2 | 8 | green ones and blue threes |
| Class 1 | 16 | 2 | 8 | blue ones and green threes |

Table 9: Query set details

