# OpenReview forum: "Attentional meta-learners for few-shot polythetic classification"
_ICLR.cc/2022/Conference — ICLR 2022 Submitted_

### Official Review · Reviewer_Hoh6 · 2021-11-01

**Correctness:** 3
**Technical Novelty And Significance:** 2
**Empirical Novelty And Significance:** 3
**Recommendation:** 6
**Confidence:** 4

**Main Review:**

Strengths:

(1) The explanation about the challenges of threshold (Prototypical networks) and attention classifiers (Matching network) is interesting.

(2) The motivation on the polythetic classification is well-motivated.

(3) Well-written and easy to follow.


Weaknesses:

(1) Lack of technological innovation. The proposed feature-selection mechanism is too simple and the only technological innovation.
They only use self-attention to obtain a better representation rather than directly using average pooling (Prototypical networks) or all the support data (matching network).

(2) How to choose repetitions R?  It is a hyper-parameter, is the bigger the R the better? Does it have no upper bound?

(3) Lack the experiments for comparison with threshold classifiers and attentional classifiers in the main paper. It is crucial to show the problems of threshold classifiers and attentional classifiers.

(4)  Self-attention also has some parameters for the transformation, why is the proposed method non-parametric?

(5) The related work is too little. Maybe consider adding some self-attention work?




**Summary Of The Paper:**

This paper first considers the limitations of threshold and attentional classifiers. They proposed an attention-based method for feature selection to address the problems of threshold classifiers and attentional classifiers.  The experiments on  several synthetic and real-world few-shot learning tasks seem good.

**Summary Of The Review:**

The implementation details and related work of the proposed paper should be further added.

---

> ### Author Response · Authors · 2021-11-17
> **Response to reviewer Hoh6**
>
> We thank the reviewer for finding that our experiments are good, that polythetic classification is well motivated, and that the paper is easy to follow.
>
> > 1. The reviewer thinks that the proposed feature selection is too simple.
>
> Response:
> - We appreciate the comment. We think that the ‘simple’ extensions we propose are a strength rather than a weakness - our approach can be easily used on top of pre-trained models to improve their performance in polythetic tasks. This is highly valuable. From our perspective, it is not clear how (or why) a more complex method would translate into better results or usability. If the reviewer has more specific suggestions in this regard they will be highly appreciated.
>
> > 2. The reviewer wonders how to properly choose the number of self-attention iterations and whether this hyperparameter has an upper bound.
>
> Response:
> - R is indeed a hyperparameter and, as such, can be optimised on held-out data. We study the behaviour of the number of self-attention iterations R. Figure 4 shows that a large enough number of iterations leads to active features identifying themselves.
> The hyperparameter has no upper bound, with higher values of R leading to sharper feature vectors. To further show the convergence of the feature vectors across the self-attention iterations, we have added a new figure in Appendix E, which we hope the reviewer finds illustrative of the feature selection process.
>
> > 3. The reviewer is concerned about the lack of comparison between threshold classifiers and attentional classifiers.
>
> Response:
> - The comparison between threshold and attentional classifier is the main theme of the paper. In section 3 (challenges in meta-learning polythetic classifications), we discuss the shortcomings of threshold classifiers for meta-learning polythetic functions. We later propose attentional classifiers as an alternative and analyse their behaviour in the polythetic setting. We characterise several properties of threshold and attention classifiers in figures 2, 3, and 5.
> - In Section 5 (experiments), we compare the performance of threshold meta-learners (e.g. protonets) with attentional classifiers (e.g. matching networks) under a broad range of datasets, including the synthetic binary strings and polythetic MNIST datasets and the real Omniglot and TieredImagenet datasets.
> - In summary, we believe that the comparison of threshold and attentional classifiers has been covered in great detail throughout the paper - if the reviewer has any concrete suggestions on how this comparison can be further strengthened, we will take them into account.
>
> > 4. The reviewer wonders why the proposed method is non-parametric.
>
> Response:
> - As presented in Algorithm 1 (Section 4), the feature selection method is non-parametric because there aren’t any weights involved in the self-attention equation $\mathbf{X}_k \leftarrow \text{softmax}(\tau\mathbf{X}_k\mathbf{X}_k^T)\mathbf{X}_k$ that updates the feature vectors $\mathbf{X}_k$ independently for each class $k$ (lines 3 and 4). Note that the temperature $\tau$ is a non-trainable hyperparameter.
>
> > 5. The reviewer thinks that the related work section is too short.
>
> Response:
> - Section 6 outlines work that is very closely related to the tasks and methods studied in this paper. Other work that helps to further motivate this study is also discussed in Section 2 (in addition to the related work section). That said, we will extend Section 6 with other approaches successfully applied for few-shot learning tasks. We will include a discussion and experiments on MAML which hopefully will strengthen this section. We would also appreciate specific suggestions of work that is highly relevant to our paper, but was overlooked.
>
> > 6. The implementation details are not available.
>
> Response:
> - We will include a ‘Reproducibility statement’ in the updated manuscript. The code is available as a supplementary, since our initial submission. We also discuss specific details regarding the experimental setup (and other models’ parameters) in Appendices E, F, H, I and J.

---

> > ### Comment · Reviewer_Hoh6 · 2021-11-30
> > **Reply**
> >
> > I would like to thank the authors for answering all my questions. I will update my initial rating to 6.

---

> ### Author Response · Authors · 2021-11-28
> **Revision**
>
> We have posted a revision addressing the concerns the reviewers collectively highlighted. We hope the reviewer finds our initial comments and this revision to answer their concerns. If there are any further concerns we would appreciate hearing them whilst we are able to respond and if not hope the review finds our revision satisfactory. Thanks again for taking the time to review our work.

---

### Official Review · Reviewer_q98z · 2021-11-02

**Correctness:** 3
**Technical Novelty And Significance:** 3
**Empirical Novelty And Significance:** 3
**Recommendation:** 6
**Confidence:** 4

**Main Review:**

Pros:
1.	Strong motivation and introduction of monothetic and polythetic classifications, drawing from prior work from other fields
2.	Interesting analysis and excellent visualizations
3.	Uses simple Boolean tasks to explain background conceptually
4.	Well-designed toy datasets and experiments to verify hypotheses of the nature of ProtoNets and Matching Networks in the context of the paper
5.	Well-written

Cons:
1.	Practical value isn’t clearly demonstrated; improvements over matching networks for the only non-toy dataset (TieredImageNet) is marginal at best.
2.	Limited baselines in experiments

More detailed comments:

This paper examines few-shot learning in the context of monothetic and polythetic classifications. This perspective sheds some light on how ProtoNets and Matching Networks perform classification of features, and there is some interesting analysis to back these hypotheses up. Concepts from the paper are well-explained with several simple examples, and toy settings are designed to illustrate them empirically. Furthermore, the analysis leads to a simple self-attention-based solution to improve features for classification. Overall, this paper reads well, and the analysis leads to some interesting insights into few-shot learning.


Empirically, the results are a little more disappointing. While several toy experiments (e.g. XOR binary strings, polythetic MNIST) are well-designed to illustrate the advantages of considering polythetic patterns, it’s not as clear how much the real-world exhibits these characteristics. The proposed method strongly outperforms the baselines in the toy settings, but improvements over the Matching Networks baseline on TieredImageNet is marginal at best. As such, as nice as these insights are, it’s not clear if it’s practically useful. Additionally, while I understand the focus was comparing with ProtoNets and Matching Networks as representatives of threshold-based and attentional classifiers, it would have been nice to have included more baselines in the experiments.


Questions:
1.	While conceptually easy-to-understand, I was previously unaware of categorizing classifiers as either attentional or threshold-based. Are there any other kinds, or must a non-attentional classifier be threshold based (and vice versa)?
2.	How do other few-shot methods fit into this taxonomy? For example, what are kinds of features (monothetic vs polythetic) are optimization-based methods (e.g. MAML) learning? What about simply training a classifier, or SVM?



=====Post-Discussion=====

I thank the authors for all their effort during the discussion phase to clarify various questions about their paper and for running additional baselines. After reading the other reviews and seeing the authors' responses, my recommendation remains mostly the same. The new perspective on meta-learning provided by the authors is an interesting one, but the practical benefits of this approach can still be more concretely demonstrated. Additional experiments in such use cases (e.g. the DNA example mentioned in one of the discussions, if such a dataset exists) would significantly strengthen this paper.

**Summary Of The Paper:**

This paper discusses monothetic and polythetic classifications in the context of few-shot learning: distinguishing similar classes often require reasoning with combinations of certain features, which can be especially challenging when only a few training examples are available. Two common types of few-shot methods, ProtoNets and Matching Networks, are shown to be threshold classifiers and attentional classifiers, respectively, each with their advantages and disadvantages. The authors introduce a simple method based on self-attention as a solution to these challenges, with experiments on several toy tasks and the more real-world TieredImageNet.

**Summary Of The Review:**

As stated in the main reviews, the empirical results aren’t particularly impressive, but overall the paper does provide some interesting analysis that may encourage new ways of thinking about few-shot learning problems. As such, I think this paper may be of interest to the ICLR community.

---

> ### Author Response · Authors · 2021-11-17
> **Response to reviewer q98z**
>
> We thank the reviewer for the positive and constructive review. We are encouraged to hear that they found our work well-motivated, leading to interesting insights into few-shot learning.
>
> > 1. The reviewer believes that the practical value of our approach is not demonstrated.
>
> Response:
> - Our research investigates meta-learning from the perspective of polythetic classification, i.e. in addition to generalising over unseen classes (most few-shot learning studies fall into this category), our work studies generalisation over unseen ways of categorising. This perspective is quite unique and, to our knowledge, it has not received much attention so far. Our hope is that our work will inspire new ways of thinking about meta-learning and expand the horizons on what can be achieved.
> - There is a lack of few-shot learning datasets where the goal is to generalise over ways of categorising rather than over unseen classes (with Omniglot alphabets and tieredImagenet being exceptions), which is why we introduced a novel dataset (polythetic MNIST) along with the binary strings tasks to further highlight the capabilities of polythetic meta-learners.
>
> > 2. The reviewer thinks that the baselines are limited.
>
> Response:
> - To address the reviewers’ point, we will include additional experiments with MAML (in the form of a neural network with a threshold classifier) to study its ability to generalise to unseen ways of categorising and deal with task-irrelevant features. We have not focused on MAML so far because the approach is model-agnostic and thus we cannot characterise this method in terms of threshold/non-threshold classifiers (i.e. it can be applied to a base model with either classifier). We are currently carrying out new experiments with MAML and preliminary results show that the MAML accuracy on the Omniglot ‘characters’ task (20-way, 5-shot) is $94.0 \pm 0.08$, while the accuracy on the ‘alphabets’ task is $89.9 \pm 0.28$. Note that these results have been obtained under the same experimental settings as for the other baselines reported in the paper.  We hope that the inclusion of the MAML baseline will strengthen our experimental section.
>
> > 3. Are there any other kinds of classifiers, other than threshold and attentional classifiers? How do other few-shot learning methods fit into this taxonomy? What kinds of features are optimisation-based methods (e.g. MAML) learning?
>
> Response:
> - This is a sensible question and something we also considered. While threshold (e.g. prototypical networks) and attentional (e.g. matching networks) classifiers encompass a wide range of methods, there are some methods that do not clearly fit into these categories (e.g. MAML). There exist other methods such as Infinite Mixture Prototypes (discussed in the paper) that lie somewhere in between the two types of classifiers -- they represent each class by a set of prototypes and can therefore fit more complex data distributions than prototypical networks.
> - Optimisation-based methods such as MAML do not fit into this taxonomy - in particular, MAML is model-agnostic and thus the approach cannot be classified in terms of threshold/non-threshold classifiers. When the MAML base model is a neural network with a final dense layer to compute class probabilities (most common form), the resulting model belongs to the category of threshold-based classifiers. Yet, in the fine-tuning stage, they might still be able to adapt to polythetic tasks by learning pseudo-variables from the support set. We believe this is an interesting point and we will further include a discussion on the taxonomy of few-shot learning methods in the main manuscript, with a focus on MAML.

---

> > ### Comment · Reviewer_q98z · 2021-11-19
> > **Reply**
> >
> > 1. On practicality
> >
> > I agree with the authors that analyzing few-shot learning in the context of polythetic classification is an interesting perspective. It was my intention to state this in my initial review; apologies if that didn't come across. My concern was more that it isn't empirically demonstrated why this perspective is useful. The authors create new datasets to illustrate the effectiveness of their solution; on the one hand, filling a need is a nice contribution, but unless it's actually a need and grounded in something real, it's just inventing toy problems for the purpose of solving them. Again, the improvements over the baseline for the most realistic dataset considered (TieredImageNet*) is marginal at best and well within the error bars. I think this paper could be significantly improved if it could demonstrate concrete value on a more real-world application than XOR, binary strings, and MNIST.
> >
> > *On Reviewer f8L4's comment on TieredImageNet: If the ResNet-18 used in the experiments was indeed pre-trained on the whole of ImageNet, then I also have concerns about the validity of the results, and not only from the test images possibly appearing in the feature extractor's training set. A major goal of few-shot learning is to generalize a model to previously unseen classes with few samples. If the model was pre-trained on the whole of ImageNet, then the novel classes aren't truly previously unseen.
> >
> > 2. Limited baselines
> >
> > While I think MAML would be a welcome addition, just adding MAML alone wouldn't solve the issues of having limited baselines. There are other strategies, some of which I think could also fit into the attentional/threshold dichotomy (e.g. nearest neighbors, RelationNet).

---

> > > ### Author Response · Authors · 2021-11-28
> > > **Revision**
> > >
> > > We have posted a revision to the manuscript that rectifies the experimental setting for the tieredImageNet task. As the reviewer correctly points out, this task is the most realistic problem we cover and in the corrected experimental setting we find our method produces far more convincing improvements.
> > >
> > > We also took the reviewer’s suggestion to compare more widely with other meta-learning approaches to alleviate the problem of limited baselines.
> > >
> > > We hope these amendments satisfy the reviewer. We would like to thank the reviewer again for their review and interaction in the rebuttal phase that we feel has help to improve our work.

---

### Official Review · Reviewer_f8L4 · 2021-11-03

**Correctness:** 3
**Technical Novelty And Significance:** 3
**Empirical Novelty And Significance:** 2
**Recommendation:** 6
**Confidence:** 4

**Main Review:**

STRENGTHS:

The few-shot polythetic classification problem is interesting and novel, and the shortcomings of baseline approaches are readily apparent. Work is well-grounded in older prior literature and the XOR_alpha task is extensively analyzed both theoretically and empirically. Writing is clear and concise (though not always easy to understand, see below). The proposed method performs well in its intended setting.

WEAKNESSES/ISSUES (in no particular order):

- The XOR_alpha meta-learning task is never formally described (it is only described as a function of bits on page 3), and so subsequent sections become difficult to follow. Please state clearly in section 2 that XOR_alpha also refers to the collection of (presumably?) all alpha-variable XOR “tasks” in a given binary n-space with n>=alpha, with a (presumably?) random partition of train and test tasks. Perhaps also consider using different notation when referring to the meta-learning problem (i.e. caption for Fig.3 right) vs the task (i.e. second to last paragraph of pg.4). Either way, the problem setup for this and other experimental tasks more broadly (i.e. the non-standard tiered-ImageNet task) should be elaborated up front and more clearly. As is I would have a difficult time re-implementing some of these benchmarks without the provided code.
- Contrary to claims in the abstract and conclusion, the paper does not show that the embedding space of a threshold classifier must grow exponentially with the number of features. Pg.4 discusses embedding growth but has it at O(n^alpha), which for given alpha is only polynomial in the number of features. The embedding space does grow exponentially with task complexity alpha, which is itself O(n), but they are not the same and these claims should be clarified (assuming that this is in fact authors’ intended argument).
- On a much broader level, I have a hard time understanding the motivation for this work. What is the envisioned use case scenario for this kind of approach? The synthetic examples, while interesting, are very contrived (which is fine since they’re synthetic) but the paper does not provide a strong motivation for the “real world” examples (omniglot and tiered-ImageNet) either. These are also somewhat contrived: while the fine-to-coarse generalization task under discussion is interesting, it’s the opposite coarse-to-fine generalization that has a clear use case. What is the envisioned use-case for a polythetic meta-learner? The introduction suggests certain kinds of fine-grained classification, but if this is the case then it should be investigated directly, such as with CUB or tiered meta-iNat benchmarks.
- The first paragraph of page 2 is difficult to parse, both because sentences are somewhat long and mostly because it is not immediately clear how conclusions are following from premises (i.e. the described 45degree rotation/”change of basis” of the OR function was initially confusing to me because this geometric operation does not translate intuitively into pure Boolean algebra; the rates of growth (2^2^n and 2^n^2) are not immediately obvious). This section could use some elaboration and clarification.
- I do not find Appendix A convincing. Appendix A claims to be a demonstration that protonets do not generalize to unseen variable combinations. While it is true that train and test combinations do not overlap, by my understanding the chance performance here is clearly due to the fact that training time noise features are test time signal features, and vice versa, and the network has simply (and correctly!) learned to suppress the “relevant” noise features. As such, no learning-based classifier should be able to solve this problem as presented; it is less a demonstration of the limits of protonets and more a demonstration of the no free lunch theorem. A much more useful and interesting set of results would be to partition the combinations of active variables randomly over train and test, so that combinations do not overlap but features are equally active in expectation at both train and test time. By my understanding this is exactly the experiment in Fig.3 right, though, so perhaps this ought to just be removed entirely.
- Authors use a pre-trained ResNet-18 for the tiered-ImageNet experiment, but the provided source for the model is PyTorch itself. How was this model pretrained? If the model was pretrained on ImageNet, then it has been trained on the test images and these results are not valid.
- Typo in Algorithm 1 Input: the an arbitrarily ordered matrix -> an arbitrarily ordered matrix


**Summary Of The Paper:**

Authors propose the general problem of few-shot polythetic classification, where class membership is determined by the combinations of present and absent features and the salient features and combination patterns change at test time. Authors demonstrate that prototypical approaches with linear decision boundaries respond poorly to these highly non-linear problems, while attention-based soft nearest neighbors paradigms work well but overfit to specious cues. An attention-based feature refinement technique is proposed and evaluated, beating both baselines on specifically polythetic few-shot tasks.

**Summary Of The Review:**

The paper proposes an interesting problem, and based on theoretical and empirical analysis provides a neat solution. Issues stem from the perhaps overly concise writing; many aspects of the paper are in need of elaboration. These include the motivation, problem setups for the various benchmarks, and the reasoning behind certain conclusions. Currently I am scoring the paper as if my understanding is correct and authors’ intended arguments are simply wrong, but if these issues can be clarified I’d be happy to raise my score.

POST-DISCUSSION:

Authors mostly address the above issues, in discussion and in the revised manuscript. I have some lingering concerns, shared with reviewer q98z, regarding the lack of _demonstrated_ practical benefit, and the fact that accuracy gains in the more “real-world” benchmarks shrink substantially relative to the motivating XOR problem. However, the revised paper is much improved, conceptual novelty remains high, and on the whole the paper is of sufficient quality for acceptance.

---

> ### Author Response · Authors · 2021-11-17
> **Response to reviewer f8L4**
>
> We thank the reviewer for their insightful feedback and for finding our solution neat and our work well-grounded.
>
> > 1. The reviewer recommends presenting the problem setup of the $XOR_{\alpha}$ upfront (as well as for the tiered-ImageNet task) and in a more clear way, with the aim of making subsequent sections easier to follow.
>
> Response:
> - The reviewer’s understanding of the $XOR_{\alpha}$ task is correct. We will clarify in Section 2 that $XOR_{\alpha}$ comprises the set of tasks on an n-dimensional binary space where labels are produced by computing the XOR over $\alpha \le n$ random bits. The $XOR_{\alpha}$ meta-learning problem (caption of Figure 3, right) and the task (second to last paragraph of page 4) are exactly the same - we will improve the caption and broadly present the task in advance to avoid any confusions.
> We will elaborate on all the experimental tasks and ensure that they can be easily understood by readers of different backgrounds.
>
> > 2. About the claim in the abstract and conclusion regarding the exponential growth of the embedding space.
>
> Response:
> - Thank you for highlighting this error in the text of the abstract and conclusion, we meant to state that the embedding space of a threshold classifier grows exponentially in the number of _active_ features, $\alpha$, as discussed in Section 3, and we will clarify in the manuscript that the embedding dimension is polynomial in the total number of features, $n$, for a fixed number of active features.
>
> > 3. The reviewer is concerned that the paper does not provide a strong motivation for real world examples where our polythetic meta-learner has a clear use case. What is the envisioned use-case for a polythetic meta-learner?
>
> Response:
> - Our research investigates meta-learning from the perspective of polythetic classification, i.e. in addition to generalising over unseen classes (most commonly addressed in most few-shot learning studies), our work studies generalisation over unseen ways of categorising. This perspective is quite unique and, to our knowledge, it has not received much attention so far. Our hope is that our work will inspire new approaches for meta-learning, leading to real-world applications in domains such as biology, medicine and healthcare, where data is often scarce and classifications are typically more nuanced.
> - For instance, we envision that such polythetic meta-learners might be useful to classify rare diseases from DNA sequences (where each sample corresponds to a long sequence of 4 possible nucleotides and mutations often lead to different phenotypes). There exist about 7000 rare diseases (affecting ~1/17 of the worldwide population) for which data is extremely limited and complex. In such scenarios, few-shot learning approaches able to genelise over unseen combinations of mutations (i.e. ways of categorising), in a similar vein as our binary strings experiment, may lead to better performance in diagnosing rare diseases. This might additionally shed new insights into their molecular mechanisms.
> - There is currently a lack of inherently polythetic few-shot learning datasets (Omniglot alphabets and tieredImagenet being exceptions). This is why we introduced a novel dataset, polythetic MNIST, along with the binary strings tasks to further highlight the capabilities of polythetic meta-learners, but we argue that this reflects that existing datasets have been developed to test existing methods and is not good evidence of the kinds of tasks that exist in the world.

---

> > ### Author Response · Authors · 2021-11-17
> > **Continuation of response to reviewer f8L4**
> >
> > > 4. The reviewer finds that, in the first paragraph of page 2, it is not clear how the change of basis of the OR function translates into Boolean algebra, and how the rates of growth ($2^{2^n}$ and $2^{n^2}$) are derived.
> >
> > Response:
> > - Our intention with the OR example was to show how threshold functions may appear polythetic due to the choice of basis, e.g. $(x, y)$, but not in another, e.g. $(x+y, x-y)$, where OR is a unary function of $(x+y)$. We concede that there is some incompatibility here between Boolean algebra and linear transformations which is why we wrote ‘rotated’ originally, but we are happy to act on the reviewer’s feedback and make this section more exact. We can perhaps write more clearly that binary $OR(x,y)$ is equivalent to the unary $MIN(x+y, 1)$, which is in the 1-d basis $(x+y)$? The main idea here is that for all linear threshold functions, there exists a linear transformation (i.e. $\mathbf{x}' = \mathbf{A}\mathbf{x}$, where $\mathbf{x}'$ and $\mathbf{x}$ are the new and old basis, respectively) such that the function is unary in the new feature basis (and thus monothetic).
> > - Regarding the rates of growth, $2^{2^n}$ is given as follows: There are $2^n$ binary strings of length $n$. A Boolean function of n-variables is defined by how it labels each of these $2^n$ strings. There are thus $2^{2^n}$ Boolean functions of $n$-variables. The bound on threshold functions is not so straightforwardly derived and is the subject of the reference to Anwar A Irmatov, _On the number of threshold functions_, 1993.
> > - Our intention is to elaborate and clarify this section as we have here, is this acceptable to the reviewer?
> >
> > > 5. The reviewer argues that no learning-based classifier can generalise on combinations of components that were inactive at train time. The reviewer also suggests partitioning the combinations of active variables randomly over the train and test sets to assess whether protonets are able to generalise across new ways of categorising.
> >
> > Response:
> > - We agree with the reviewer on why protonets might fail to generalise across components that are inactive at train time - the network learns to suppress them because they are not informative of class labels. Nonetheless, we think that this is not necessarily the case for other learning algorithms - our approach is precisely well-suited for generalising across unseen ways of categorising (as demonstrated in other experiments, e.g. classification of Omniglot alphabets). In other words, given a set of informative features (extracted for instance via unsupervised learning), the proposed feature selection mechanism adaptively dilutes task-irrelevant features even for tasks that are driven by previously unseen combinations of features. We acknowledge that training the entire model in an end-to-end manner is problematic and we have included this experiment for completeness. We will make clear in the Appendix that the result we present is exactly as expected.
> > - The reviewer’s idea on randomly splitting combinations of active features over the train and test sets is an improvement with respect to our initial experiment. This is different from what we did in Figure 3 _right_ (there, all combinations of active features are seen at train time). We will conduct the proposed experiment, which we expect to reinforce the claim that protonets struggle to generalise across new ways of categorising.
> >
> > > 6. Results on tiered-Imagenet are invalid.
> > - We are currently investigating this. In the case the pre-trained network was trained on images that belong to the test set of tiered-Imagenet, we will repeat our experiments using otherwise pre-trained models and update the manuscript accordingly.
> >
> > > 7. Typo in the algorithm.
> > - We thank the reviewer for spotting the typo. This will be amended in the updated manuscript.

---

> > > ### Comment · Reviewer_f8L4 · 2021-11-20
> > > **Reply**
> > >
> > > My thanks to the authors for the response and clarification, this is very helpful.
> > >
> > > Regarding 1: there appears to be some overloading of the word “task” here. In my review I meant “task” to be synonymous with “episode”, i.e. a single choice of parity function, as introduced in pg3, and “problem” to refer to the demand of generalizing from train to test “tasks”. My recommendation was that this distinction be made formal (so that contrary to author response, it would be impossible, by definition, for the “meta-learning problem” and the “task” to be “exactly the same”). But this appears to just be an unfortunate word choice issue, otherwise we appear to be in agreement and no further discussion is needed.
> > >
> > > Regarding 3: the DNA sequencing example is an excellent motivating example and its addition would greatly strengthen the paper. Responses to reviewers PpwW and q98z also stress the conceptual novelty of polythetic few-shot classification over the practical novelty, and for the ICLR venue this is not inappropriate. It also occurs to me that coarse-to-fine few-shot learning, in addition to the provided fine-to-coarse setting, might also be an example of polythetic meta-learning. As such, reversing the train/test predictions on the omniglot/tiered-ImageNet experiments (so train on supercategories, test on subcategories, instead of vice-versa) could greatly strengthen these results and demonstrate immediate practicality (assuming current results hold). Unfortunately that’s likely out of scope at this point; I apologize for not bringing this up earlier in the review process.
> > >
> > > Regarding 4: thanks for the clarification, this is indeed much better.
> > >
> > > Regarding 5: I see, I had not appreciated this motivation for the experiment. It could be interesting to see whose intuition is correct – are noise features suppressed during training beyond the ability of the self-attention mechanism to recover at test time? – by comparing protonet performance to matching and self-attention matching networks directly on this problem. These results would greatly strengthen the experiment regardless of exact setup.
> > >
> > > In sum, most of my concerns are addressed and I will be raising my score. I have held off on doing so pending the results of the investigation into tiered-ImageNet pretraining (and/or the results of the new experiment from point 5, should authors wish to report them, though like the authors I expect the results for that one are fairly obvious).

---

> > > > ### Author Response · Authors · 2021-11-28
> > > > **Revision**
> > > >
> > > > We have posted a revised manuscript that incorporates the advised changes and presents new (much improved) results on the tieredImageNet task. We hope the reviewer finds that the changes suitably address their concerns.
> > > >
> > > > We would like to thank the reviewer again for their constructive reviewing that we feel has significantly improved our work.

---

### Official Review · Reviewer_PpwW · 2021-11-03

**Correctness:** 3
**Technical Novelty And Significance:** 3
**Empirical Novelty And Significance:** 4
**Recommendation:** 6
**Confidence:** 3

**Main Review:**

**Strengths**

This paper discusses meta-learner from a very unique perspective, with the shortcomings of both ProtoNet-liked threshold meta-learners and MatchingNet-liked attentional meta-learner discussed solidly with intuitive examples and convincing derivation of misclassification rates.

The proposed attentional feature selection is simple yet effective, and can potentially guide and stimulate many follow-up improvements to meta-learning.

**Weaknesses**

My main concern is that this paper tends to isolate itself from the entire research of meta-learning, and focus on a corner instead.
ProtoNets (representing threshold meta-learners) and MatchingNets (representing attentive meta-learners) are insufficient to cover the entire research of meta learning. There are many other directions of meta-learning that are completely overlooked in the discussion.
The most representative case is the inner-loop optimization-based meta-learner like MAML [1], where the inner-loop adaptation to the feature extractor can potentially solve the challenge of 'not all features are relevant in all tasks and that the support is unlikely to span the input domain' pointed out in this paper.

And many other directions and methods, e.g., methods based on Hebbian rules [2] might share a similar spirit with the proposed method, and the presentation can be further improved by incorporating more comprehensive discussion to the latest progress of meta-learning and the connections of the proposed method to others.

**Minor**

The experimental settings included in this paper seem a bit weak. More common benchmarks on real-world meta-learning (classification) tasks, like miniImageNet, and the challenging cross-domain settings can better support the discussions.

The overall writing is good, but some further improvements are expected. For example, is the very first sentence of this paper grammatically incorrect? I believe it's better to say 'that need to be neither universal nor ...'

[1] Model-Agnostic Meta-Learning for Fast Adaptation of Deep Networks, ICML2017

[2] Differentiable plasticity: training plastic neural networks with backpropagation, ICML 2018

**Summary Of The Paper:**

In this paper, the authors present a discussion on two main-stream meta-leaners with attentional classifiers and threshold meta-learners under a unique view of polythetic classification. And to address the limitations of both, attention-based feature selection is introduced.
Improved performance is demonstrated by both synthetic and real-world few-shot learning tasks.


**Summary Of The Review:**

I give an initial recommendation of score 6 mainly for appreciating this novel view to mainstream meta-learners, and solid discussions supporting the points. However, to meet and standard of ICLR and demonstrate a clear contribution to the research of meta-learning, I believe further efforts on comprehensive discussions are highly expected.

---

> ### Author Response · Authors · 2021-11-17
> **Response to reviewer PpwW**
>
> We thank the reviewer for the positive and constructive review. We are encouraged to hear that they found our perspective very unique, that the examples are intuitive, and that our work can stimulate many improvements to meta-learning.
>
> > 1. The reviewer is concerned that our research is isolated and does not cover the entire meta-learning field, e.g. MAML, are overlooked in the discussion. The reviewer suggests that other methods (e.g. based on Hebbian rules, such as differential plasticity) could be included in the discussion.
>
> Response:
> - Indeed, we agree that research on meta-learning, and in particular for few-shot classification, is much broader. Our research investigates meta-learning from the perspective of polythetic classification, i.e. in addition to generalising over unseen classes (most few-shot learning studies fall into this category), our work studies generalisation over unseen ways of categorising. This perspective is quite unique and, to our knowledge, it has not received much attention so far. In our opinion, this is highly valuable because, as the reviewer points out, our work can shed new insight into meta-learning and catalyse new developments.
> - As such, this work aligns more closely with meta-learning approaches for metric learning. In particular we mainly focus on evaluating the performance of prototypical networks and matching networks as representatives of threshold/attentional approaches, respectively. In the paper, we discuss the benefits and limitations of these two paradigms mostly through the lens of the two representatives, but also highlight other recent methods (in Section 6) that are closely related to these paradigms.
> - We did not focus on MAML because it does not fit into our taxonomy - MAML is an optimisation-based method that is model-agnostic and thus the whole approach cannot be classified in terms of threshold/non-threshold classifiers. Nonetheless, we believe the reviewer’s suggestions are sensible and we will include additional experiments with MAML (in the form of a neural network with a threshold classifier). We are currently carrying out new experiments and preliminary results show that the MAML accuracy on the Omniglot ‘characters’ task (20-way, 5-shot) is $94.0 \pm 0.08$, while the accuracy on the ‘alphabets’ task is $89.9 \pm 0.28$. Note that these results have been obtained under the same experimental settings as for the other baselines reported in the paper.  We hope that the inclusion of the MAML baseline will strengthen our experimental section.
> - We thank the reviewer for the pointer on differentiable plasticity. We plan to improve the discussion in the related work section by accommodating these related approaches (as well as MAML, as discussed above).
>
> > 2. More common benchmarks on real-world meta-learning tasks, e.g. miniImageNet, can better support the discussions.
>
> Response:
>
> - We appreciate the reviewer’s comment. We believe our experimental section is very comprehensive - the tieredImageNet dataset used in this paper not only has a polythetic nature (i.e. classes are grouped into categories corresponding to higher-level nodes in the ImageNet hierarchy), but it is also bigger, newer, and more complex than miniImageNet.
>
> > 3. Is the very first sentence of this paper grammatically incorrect?
>
> Response:
> - The first sentence (i.e. “need neither be A nor B”) is grammatically correct though we are happy to rewrite the sentence if the reviewer considers it will improve the readability of the abstract. We thank the reviewer for checking this.

---

> ### Author Response · Authors · 2021-11-28
> **Revision**
>
> We have posted a revision that includes a revised related work section that we believe better situates our work within the wider field of meta-learning. We have also compared to a larger suite of baselines and improved the tieredImageNet experiments on the advice of the other reviewers.
>
> We hope that these revisions satisfy the reviewer. Thanks again for the review, particularly the pointers on the related work.

---

### Author Response · Authors · 2021-11-22
**Response to all reviewers**

Once again, we thank the reviewers for their valuable feedback. After carefully considering all their comments, we have addressed the following points in the updated manuscript:
- As pointed out by reviewers f8L4 and q98z, using a ResNet model pre-trained on ImageNet as a feature extractor for tieredImageNet is likely incorrect - we were not able to ensure that there is no overlap between test images of tieredImageNet and the train images of ImageNet. We thank the reviewers again for spotting this crucial issue. **To amend this**, we have conducted new experiments with another publicly available ResNet-12 (Zhang et al., 2020) that was pre-trained on the training set (i.e. training classes and images) of tieredImageNet. We used the exact same network and evaluation settings for all competing approaches. Our new results show that the proposed feature selection mechanism (followed by an attentional classifier like MN) outperforms PN and MN on tieredImageNet in a statistically significant way (p-value < 0.001). Note that, in this new setup we attempted to also evaluate  Infinite Mixture Prototypes (IMP) but to no avail (out-of-memory issues on a 12GB GPU). Therefore we evaluated it on the Omniglot problem, as discussed next.

- To address the comments of reviewers q98z, PpwW, and Hoh6, we have conducted additional experiments with more baselines. In addition to Infinite Mixture Prototypes (IMP), for which we now evaluate performance on Omniglot, we also report results for MAML and Nearest Neighbors (using a feature extraction model trained using PN as in FS*). All methods are trained/evaluated under the same settings. We also extended the related work section to discuss how our work aligns with a broader variety of few-shot learning methods as suggested. We thank the reviewers for the pointers to related methods and literature.

- We have improved the experiment of Appendix A according to the suggestions from reviewer f8L4. Results indicate that PN fails to generalise to unseen non-threshold functions. Interestingly, their performance is slightly better-than-chance for very large embedding dimensions (i.e. for 100-dimensional embeddings) - we hypothesise that, by chance, large numbers of non-linear features (e.g. output by a random feature extractor) will include some features that are useful for the task of interest (i.e. similar to extreme learning machines).

- We have performed a number of changes in the manuscript to improve its clarity and readability. In particular, we have (among others):
    1. Improved the problem descriptions (e.g. $XOR_{\alpha}$).
    2. Amended the claims about the exponential growth of the PN embeddings.
    3. Improved the explanations on changes of basis and the number of Boolean functions.
    4. Disambiguated the terms “problem” and “task”.
    5. Extended the discussion in our related work section to include connections with other recent meta-learning approaches.
    6. Provided a short discussion on the broader impact of our work with a more tangible potential use-case example of polythetic meta-learners for diagnosing rare diseases.
    7. Added a reproducibility statement.

---
Omniglot results (as presented in Table 3 in the revision)

|   | Characters     	| Alphabets      	|
|--------|------------------------|------------------------|
| PN 	| **98.6 ± 0.0** | 83.4 ± 0.3     	|
| MN 	| 91.1 ± 0.1     	| 78.4 ± 0.3     	|
| FS+MN  | 96.2 ± 0.0     	| 94.2 ± 0.2     	|
| IMP	| **98.6 ± 0.0** | **96.0 ± 0.2** |
| MAML   | 94.0 ± 0.1     	| 89.9 ± 0.3     	|
| MN*	| 97.9 ± 0.1     	| 81.3 ± 0.3     	|
| NN*	| 98.3 ± 0.0     	| 95.7 ± 0.3     	|
| FS+MN* | 98.1 ± 0.0     	| **96.0 ± 0.2** |
| FS+NN* | 98.3 ± 0.0     	| **96.0 ± 0.2** |

---
TieredImageNet results (as presented in Tables 4 and 5 in the revision) by number of categories (C) the 'way' and number of subgroups (G) within each category.

| G  |   | C=2 | C=4 | C=8 |
|----|---|-----|-----|-----|
| 5  | FS  |**83.5 ± 0.3**|**66.4 ± 0.3**|48.9 ± 0.2|
| 5  | MN  |82.7 ± 0.4|65.4 ± 0.3|48.3 ± 0.2|
| 5  | PN  |81.4 ± 0.3|64.6 ± 0.3|**49.4 ± 0.1**|
| 10 | FS  |**83.6 ± 0.3**|**65.8 ± 0.3**|**48.7 ± 0.1**|
| 10 | MN  |82.9 ± 0.3|64.9 ± 0.3|48.0 ± 0.1|
| 10 | PN  |81.1 ± 0.3|63.3 ± 0.3|48.2 ± 0.1|

| | | |C=2| C=4| C=8|
|---|---|---|---|---|---|
| **G** | **X** | **Y** | **X / Y (tie)** | **X / Y (tie)** | **X / Y (tie)** |
| 5|FS|MN|**230** / 162 (108)|**329** / 101 (70)|**343** / 113 (44)|
| 5|FS|PN|**319** / 136 (45)|**339** / 142 (19)|239 / _247_ (14)|
|10|FS|MN|**258** / 185 (57)|**357** / 108 (35)|**413** / 68 (19)|
|10|FS|PN|**374** / 101 (25)|**396** / 99 (5)|**296** / 191 (13)|

---

### Decision · Program_Chairs · 2022-01-20

**Decision:**

Reject

**Comment:**

This paper analyzes problems of existing threshold meta-learners and attentional meta-learners for few-shot learning in polythetic classifications. The threshold meta-learners such as prototypical networks require exponential number of embedding dimensionality, and the attentional meta-learners are susceptible to misclassification. The authors proposed a simple yet effective method to address these problems, and demonstrated its effectiveness in their experiments. This paper discusses meta-learning from a very unique perspective as commented by a reviewer, and clearly explained problems of widely-used meta-learning methods. However, this paper focus on prototypical networks and matching networks even though there have been proposed many meta-learning methods. Some existing methods seem not to have the problems of prototypical networks and/or matching networks. In addition, the practical benefits of the proposed approach are not well demonstrated. Although the additional experiments in the author response addressed some concerns of the reviewers, they are not enough to demonstrate the effectiveness of the proposed method.